*A Nature Portfolio journal*

# Unveiling the impact of cGMP-dependent protein kinase of *Neospora caninum* on calcium fluxes and egress functions through quantitative phosphoproteome analysis

Xianmei Wang[1], Kun Guo[1], Zhili Shan[1], Zhu Ying[1], Zifu Zhu[1], Shiman Yang[1], Na Yang[1], Qun Liu[1,2], Lifang Wang[1] & Jing Liu [1,2] ✉

*Neospora caninum*, a pathogen associated with abortion in pregnant cattle and motor nerve disorders in dogs, poses a substantial threat. Cyclic GMP-dependent protein kinase (PKG) functions as a central signal transduction hub in apicomplexan parasites. However, PKG has not been characterized in *N. caninum*, and its targets and pathways controlled by PKG remain unknown. Using a mini auxin-inducible degron system (mAID), we knocked down PKG in *N. caninum*, demonstrating its indispensable role in tachyzoite invasion and egress from host cells. PKG promotes microneme secretion and enhances tachyzoite gliding motility by elevating intracellular $Ca^{2+}$ levels ($[Ca^{2+}]_i$). Phosphoproteomics identified 1125 proteins as potential downstream targets of PKG, showing significantly reduced phosphorylation after treatment with the PKG inhibitor MBP146-78. These proteins are involved in signal transduction, transcriptional regulation, lipid transport and metabolism, vesicle transport, and ion transport. Additionally, CACNAP, a calcium channel-associated protein that facilitates calcium influx at the plasma membrane, plays a supportive role in the egress process of *N. caninum*. These findings underscore the importance of PKG and its downstream molecules in regulating egress, offering novel insights into the mechanisms underlying the activation of $[Ca^{2+}]_i$.

N*eospora caninum*, an obligate intracellular parasite belonging to the api-complexan phylum, is associated with abortion in pregnant cattle and motor nerve disorders in dogs[1]. Neosporosis, caused by *N. caninum*, is a major contributor to abortion or neonatal death in cows worldwide[2]. This disease imposes a substantial economic burden and results in extensive financial losses[3]. *N. caninum*, as an intracellular parasitic protozoan, systematically orchestrates survival, replication, and propagation within host cell. For its persistence within the host, active invasion of the host cell and initiation of replication within the parasitophorous vacuole (PV) are imperative. When proliferating or facing external challenges, tachyzoites are triggered to egress from the host cell, seeking new cells for invasion and continue growth cycles. Sustaining this survival form depends not only on the metabolic processes of the parasite, but also on the regulation of various signaling pathways within the parasite. This orchestration ensures efficient coordination of various physiological functions[4,5]. Protein kinases play a critical role in these signaling pathways as essential molecules that modify the activity, function, subcellular localization, or regulatory features of downstream proteins[6]. Despite their significance, knowledge of protein kinases in *N. caninum* remains limited. Further investigation of these molecular components is crucial for unraveling the complex mechanisms underlying parasite growth and host interactions.

Host cell Invasion and egress are two key processes of the parasite lytic cycle for transmission between host cells that are governed by multiple signals, such as the second messenger cyclic GMP (cGMP), phosphatidic

[1]National Animal Protozoa Laboratory, College of Veterinary Medicine, China Agricultural University, Beijing, PR China. [2]National Key Laboratory of Veterinary Public Health and Safety, Key Laboratory of Animal Epidemiology of Ministry of Agriculture and Rural Affairs, College of Veterinary Medicine, China Agricultural University, Beijing, PR China. ✉e-mail: liujingvet@cau.edu.cn

acid (PA), pH and ionic composition[7–9]. Cyclic GMP-dependent protein kinase, also known as protein kinase G (PKG), is an important phosphorylated kinase activated by cGMP[10]. PKG catalyzes the transfer of the γ-phosphate group from ATP to serine/threonine residues on substrates, triggering key biological effects in eukaryotic cell signaling pathways. PKG-mediated signal transduction occurs through the phosphorylation of specific substrates. In mammalian cells, substrates of protein kinase G I (cGKI)[11] perform various cellular functions, such as regulating intracellular calcium and potassium ion concentrations, calcium sensitivity, and cytoskeleton organization. Meanwhile, substrates of cGKII are involved in chloride and sodium transport, as well as transcriptional regulation[11]. The activity of these molecules is pivotal for essential functions, including tissue contraction, protein exocytosis, as well as cell proliferation and differentiation[12,13]. In apicomplexans, *Plasmodium falciparum* PKG (PfPKG) emerges as a key regulator across crucial life cycle phases[14]. Physiological processes regulated by PKG include gametogenesis in the sexual stage[15], schizont rupture in the blood stage[16], ookinete differentiation and motility in *Plasmodium berghei*[17,18], and schizont development in the liver stage[19,20]. Notably, PKG silencing via antisense RNA in *Cryptosporidium parvum* significantly inhibited the egress of merozoites, identifying PKG as a key regulator for *Cryptosporidium* merozoite egress[21]. Moreover, Kevin M. Brown et al. pioneered the development of an auxin-inducible degron (AID) tagging system for conditional depletion of PKG in *Toxoplasma gondii*, underscoring its significance role in invasion, egress, and microneme secretion[22]. PKG has also emerged as a prominent target for combating drug resistance in apicomplexan parasites[14]. The trisubstituted pyrrole 4-[2-(4-fluorophenyl)-5-(1-methylpiperidine-4-yl)-1H pyrrol-3-yl] pyridine (Compound 1) has been biochemically and genetically validated as an anticoccidial agents and an inhibitor of parasite cGMP-dependent protein kinase in apicomplexan parasites[23–26].

Reverse genetics and pharmacological investigations have shown that PKG-regulated calcium signaling controls key life cycle events in apicomplexan parasites[27–29]. Calcium ions ($Ca^{2+}$) function as versatile second messengers, activating or modulating various cellular responses such as gene transcription, proliferation, and differentiation[30,31]. Critical life cycle processes in apicomplexan parasites, including motility, adhesion, invasion, and egress, rely on the controlled release of $Ca^{2+}$ from intracellular stores. This release elevates cytoplasmic calcium levels, triggering microneme release[31,32]. PKG and its downstream proteins play a crucial role in activating calcium signaling, with PKG generally acting upstream of calcium release[4,29,33,34].

This inference is supported by observations that treatment of parasites with the PDE inhibitor BIPPO[22,35], which likely elevates cGMP levels, activates TgPKG and triggers $Ca^{2+}$ flux before egress[36,37]. This $Ca^{2+}$ flux does not occur when TgPKG is inhibited by compound 1[38]. While signal transduction by PKG is known to occur through the phosphorylation of specific substrates, the molecular events linking PKG to intracellular $Ca^{2+}$ levels ($[Ca^{2+}]_i$) in *T. gondii* remain unknown. In *P. berghei*, phosphoinositide metabolism has been considered to link PKG to essential $Ca^{2+}$ signals at key decision points in the life cycle of *malaria* parasite[27]. It is hypothesized that inositol 1, 4, 5-trisphosphate ($IP_3$), generated by phosphoinositide phospholipase C (PI-PLC)-mediated cleavage of phosphatidylinositol 4,5-bisphosphate (PIP2)[27], opens an unidentified $IP_3$-sensitive channel, releasing $Ca^{2+}$ from organelles[39]. This PKG-dependent $Ca^{2+}$ signal triggers the release of subtilisin-like protease (PfSUB1), necessary for parasite egress, as well as the release of micronemes and exonemes[40]. Previous studies indicate that mobilization of $[Ca^{2+}]_i$ prior to egress was shown to be dependent on cGMP-dependent signaling[41]; however, the related molecules involved are still unclear, warranting further studies to unveil the mechanism of calcium signaling activation by PKG.

PKG has been identified in apicomplexan parasites such as *Plasmodium*, *Toxoplasma*, *Cryptosporidium*, and *Eimeria*, though most PKG targets and pathways it regulates remain unidentified. In addition, PKG has not been investigated in *N. caninum*. This study aims to address this gap by characterizing PKG and its downstream molecules in *N. caninum*, focusing on their role in tachyzoite egress from host cells. We demonstrated that PKG may activate $[Ca^{2+}]_i$ through a calcium channel located at the plasma membrane. Further screening and identifying phosphorylated downstream targets of NcPKG will provide an important foundation for unraveling the molecular mechanism of PKG in the egress and invasion processes.

## Results

### *N. caninum* PKG is essential for tachyzoites in vitro growth

A comprehensive database search identified a single PKG gene in *N. caninum* (ToxoDB, NCLIV_055260), which encodes a 997-amino-acid protein (Supplementary Data 1). Homology research (BLAST) revealed that NcPKG contains four tandem copies of the cyclic nucleotide-binding domain (CAP family effector domains, CAP_ED) and a catalytic domain for serine/threonine kinases (STKc_cGK) (Fig. 1a), which catalyzes the transfer of the γ-phosphoryl group from ATP to serine/threonine residues on protein substrates[42]. Efforts to generate a complete PKG knockout strain using CRISPR/Cas9 technology were unsuccessful, indicating the essential role of PKG in *N. caninum*. To investigate PKG function, we employed a conditional depletion strategy using the auxin-induced degradation factor (AID) strategy[22,43]. In this approach, a mAID-Ty tag was fused to the C-terminus of PKG (Fig. 1b) in the NcTIR1 strain (sequence information in Supplementary Data 1). Western blotting analysis of parasite lysates confirmed the expression of two bands near the predicted size of PKG (predicted molecular weight: 111 kDa) (Fig. 1c). The two isoforms of NcPKG may be expressed by alternative translation initiation like TgPKG-I, which is localized to the plasma membrane via N-acylation, governing PKG function while the smaller isoform, TgPKG-II, is cytosolic due to the absence of N-terminal acylation residues[22]. Immunofluorescence analysis (IFA) revealed that PKG was localized to the plasma membrane and cytoplasm of intracellular tachyzoites (Fig. 1d), consistent with homologous proteins in other Apicomplexa, such as *T. gondii*[22]. By utilizing auxin (IAA, 500 μM) to regulate PKG protein degradation, we observed completed degradation of the target protein after 8 h of treatment (Fig. 1c, d). The plaque assay, reflecting the comprehensive survival ability of *N. caninum* in vitro, demonstrated that the *pkg*-mAID parasite line exhibited a significantly reduced formed plaques in the presence of IAA compared to the basal NcTIR1 parasites (Fig. 1e, f). These results underscore the essential role of PKG in the in vitro growth of *N. caninum*.

### NcPKG contributes to invasion, egress, and motility

Although plaquing assays provide insights into the importance of PKG during the in vitro lytic cycle, they do not precisely pinpoint which step of the cycle is affected. To assess whether PKG degradation influences parasite proliferation, we quantified the number of parasites per vacuole for each parasite line within 30 h of invasion into host cells. No significant differences were observed (Fig. 2a).

Given that PKG functions as a signal transduction molecule, our phenotypic analysis focused on signaling-regulated processes in *N. caninum*, such as egress, invasion, and gliding motility. After 8 h of IAA treatment, parasites were collected, inoculated into cells, and their invasion ability was assessed by IFA. The counts consisted of host nuclei (blue), non-invaded parasites by SRS2 staining (green), and all parasites identified by IMC1 staining (red) (Fig. 2b). The proportion of successful invasions was notably reduced compared to untreated parasites (Fig. 2c). To explore the relationship between PKG and egress in *N. caninum*, calcium ionophore A23187[44] was used to accelerate the parasite egress from host cells. This treatment demonstrated a substantial decrease in the proportion of *N. caninum* egress in the PKG knockdown line compared to the control group (Fig. 2d). Further analysis of the proportion of natural egress in the absence of PKG revealed that PKG-expressing strains initiated substantial egress at 48 h (49.6%) and reached nearly complete egress at 72 h, causing severe damage to host cells (Fig. 2e, f). In contrast, only a small fraction (6.3%) of PKG degraded strains egressed by 86 h (Fig. 2e, f), highlighting the essential role of PKG in invasion and egress processes in *N. caninum*.

Gliding motility is crucial for parasite survival as it powers host cell invasion and egress[45]. As invasion and egress of host cells rely on active

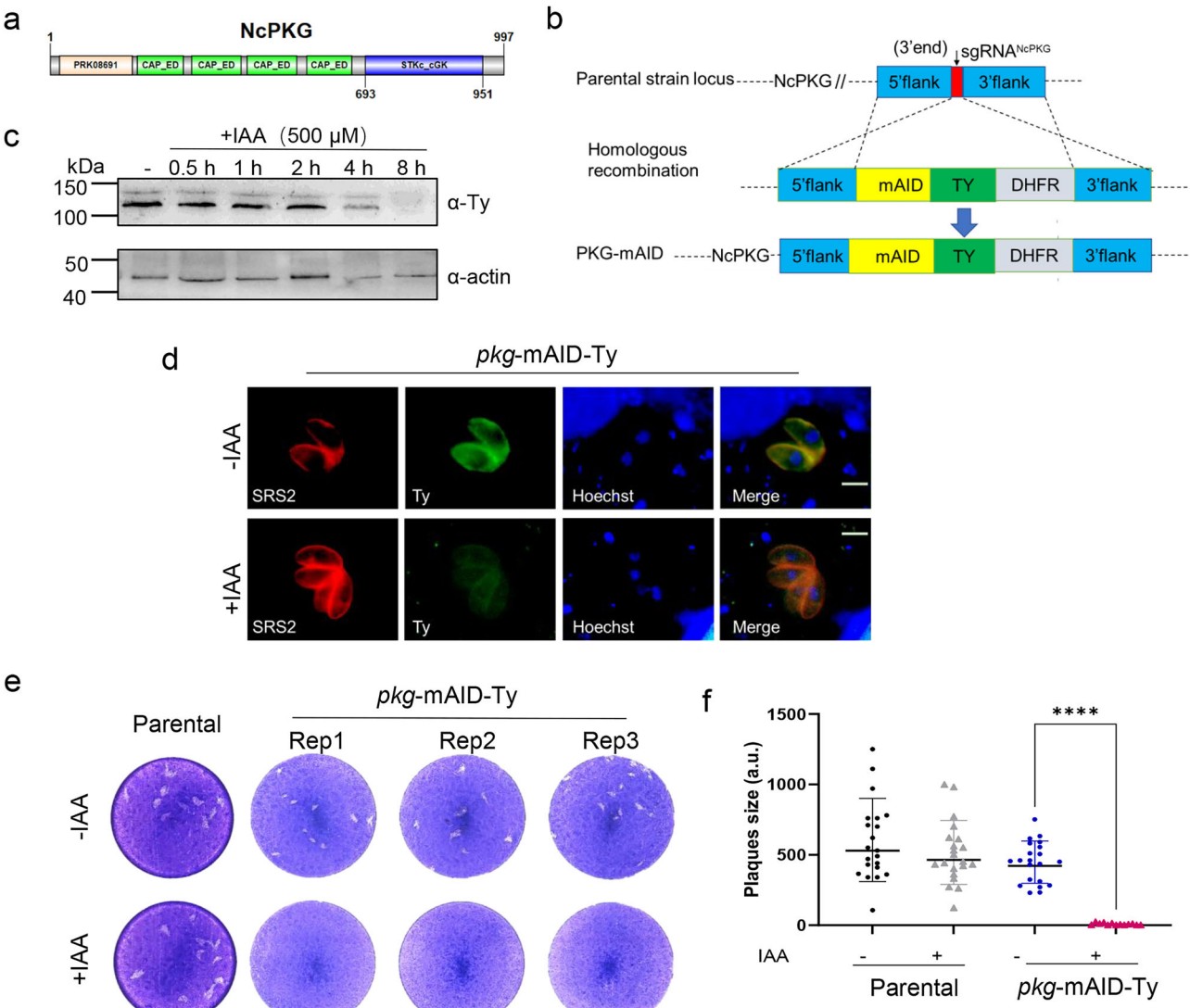

**Fig. 1 | Auxin-induced PKG protein degradation impairs the lytic cycle of *N. caninum*.** **a** Model diagram of the NcPKG domain structure, Showing four CAP family effector domains (CAP_ED) and a catalytic serine/threonine kinase (STKc_cGK) domain. **b** Diagram of the gene-specific degradation strategy using the auxin-inducible degron system. The degradation factor mini-AID (mAID) was fused to the C-terminal of PKG gene in NcTIR1 strain, generating the *pkg*-mAID strain. Ty endogenous tags were used to analyze its expression. **c** PKG content regulation by auxin for varying periods. Western blotting images showed that PKG was undetectable after 8 h of treatment with 500 μM IAA. α-Ty was used to detect PKG, and α-actin was used as a loading control. **d** IFA showing that PKG was initially located in the plasma membrane and cytoplasm of tachyzoites, with fluorescence loss after 8 h of IAA treatment. PKG was stained with α-Ty (green), the plasma membrane with α-NcSRS2 (red), and nuclei with Hoechst (blue). Scale bars: 5 μm. **e** Plaque assays revealing growth capabilities of *pkg*-mAID mutant and parental strains. In IAA-treated medium, *pkg*-mAID exhibited reduced plaques formation, while the parental strain retained normal capacity. **f** Statistical analysis of tachyzoite plaque areas in absence or presence of IAA (500 μM). Photoshop was used to calculate the plaque area, and GraphPad Prism 9.0 was used for statistical analysis. Plaques data for each strain were obtained from three independent assays. ****$p < 0.0001$; ns, not significant (unpaired two-tailed Student's *t* test).

motility, we compared gliding motility efficiency between PKG degraded and the undegraded lines. Upon addition of 8-bromo-guanosine 3′,5′-cyclic monophosphate (8-Br-cGMP), a cell membrane permeable cGMP derivative[46] that effectively binds apicomplexan PKG effectively[8,47], extracellular motility of tachyzoites was further enhanced (Fig. 2g), with a significant increase in both the proportion (Fig. 2h) and motility distance of tachyzoites (Fig. 2i). Conversely, PKG degradation resulted in fewer parasites initiating gliding motility (Fig. 2g–i), underscoring the regulatory role of PKG in motility and its subsequent impact on invasion and egress.

### Activation of NcPKG can promote microneme secretion and enhance cytosolic calcium fluctuations

Gliding motility is dependent on adhesins secreted by micronemes[48,49]. To analyzed the effect of PKG on micronemes secretion, *pkg*-mAID strains

treated with or without IAA were induced by 8-Br-cGMP simultaneously. Tachyzoites pellet and excrete-secrete antigens (ESAs) were collected to assess micronemes secretion. The results revealed a significant reduction in the levels of secreted NcMIC11 and NcMIC6 in ESAs after PKG degradation, highlighting PKG's pivotal role in the process of microneme secretion (Fig. 2j).

Recent studies have revealed that $[Ca^{2+}]_i$ can trigger the activation of multiple molecules involved in the lytic cycle of apicomplexan parasites, encompassing processes such as motor system activation and micronemes secretion in *T. gondii* and *plasmodium*[4,36,50,51]. To investigate how PKG regulates $[Ca^{2+}]_i$ activation in *N. caninum*, we generated the Nc1/GCaMP6f strain (Supplementary Fig. 1a, b), utilizing a genetically-encoded calcium sensor (GCaMP6f) known for its sensitivity to change in $[Ca^{2+}]_i$. This sensor exhibits varying green fluorescence intensity in response to fluctuations in

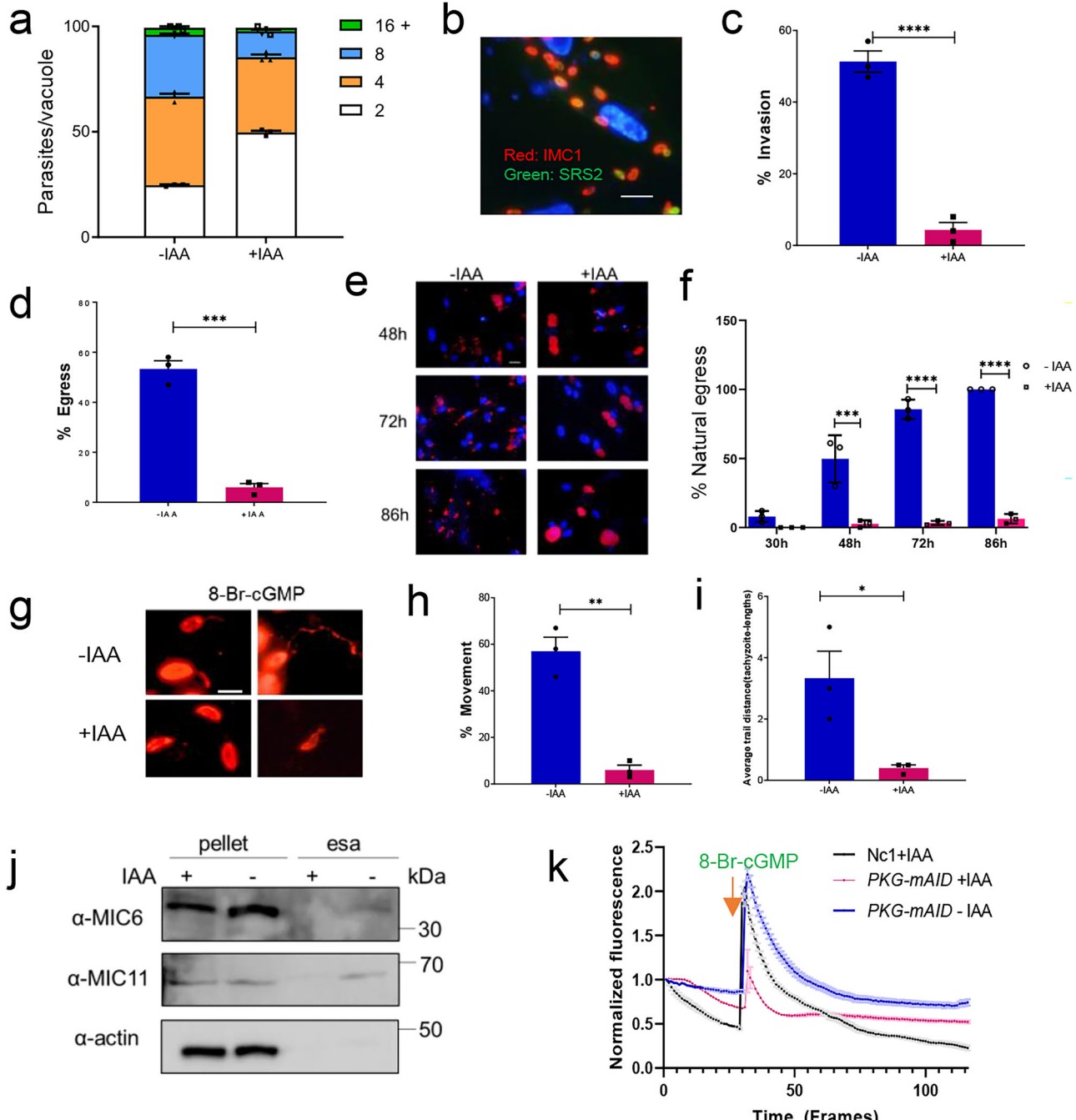

**Fig. 2 | Functional analysis of PKG in the lytic cycle of *N. caninum*. a** Comparison of intracellular proliferation in the *pkg*-mAID strain with absence or presence of IAA (500 μM) for 30 h. **b** IFA analysis of parasite invasion efficiency. Non-invading parasites were stained with α-SRS2 (green), all parasite with α-IMC1 (red), and nuclei with Hoechst (blue). Scale bars: 30 μm. **c** Quantification of invasion efficiency in *pkg*-mAID strain with the absence or presence of IAA (500 μM). Tachyzoites were precultured with or without 500 μM IAA for 8 h, then collected to invade host cells for 1 h. **d** Quantification of egress efficiency in *pkg*-mAID strain with the absence or presence of IAA (500 μM). Tachyzoites were precultured with or without 500 μM IAA for 8 h and then induced with calcium ionophore A23187 (1 μM) for 3 min. **e** Immunofluorescence images showing the natural egress of tachyzoites with the absence or presence of IAA (500 μM) at different times. Parasite plasma membrane was stained with α-NcSRS2 (red), and nuclei with Hoechst (blue). Scale bars: 50 μm. **f** Comparison of the proportion of naturally egressed parasites in *pkg*-mAID strain with absence or presence of IAA (500 μM) for varying periods. **g** Motility assay of *pkg*-mAID strain. Tachyzoites were precultured with or without 500 μM IAA for 8 h and then placed onto cell slides. Movement was induced with 5 μM 8-Br-cGMP, and parasite motion trajectory was stained with α-NcSRS2 (red). Scale bars: 5 μm. The

proportion of moving tachyzoites **h** and moving distance (tachyzoite-lengths) **i** were recorded. **j** Analysis of microneme secretion in *pkg*-mAID strain with and without IAA (500 μM). Tachyzoites were precultured with or without IAA for 8 h, then induced with 5 μM 8-Br-cGMP for 10 min. Western blotting was analyzed microneme content and secretion in pellet and supernatant ESA. Antibodies α-MIC6 and α-MIC11 were used for detection of microneme protein 6 and 11, and α-actin was used as a loading control. **k** Analysis of $Ca^{2+}$ signal activated by 8-Br-cGMP in *pkg*-mAID strain. After pretreatment with or without 500 μM IAA, cytosolic $Ca^{2+}$ changes were analyzed over time by measuring green fluorescence intensity following the addition of 5 μM 8-Br-cGMP. Time-series images were converted to video and imported into ImageJ software. Thirty parasite regions were selected, and GFP channel fluorescence intensity was quantified at each time point. GraphPad Prism was used to plot the fluorescence intensity of the calcium indicator strain over time. Unless otherwise specified, statistical analysis of the experiments was performed using GraphPad Prism 9.0. Results are presented as mean ± standard error of the mean from three experiments Significance level: *$p < 0.05$, **$p < 0.01$, ***$p < 0.001$, ****$p < 0.0001$, ns, not significant (un*p*aired two-tailed Student's *t* test).

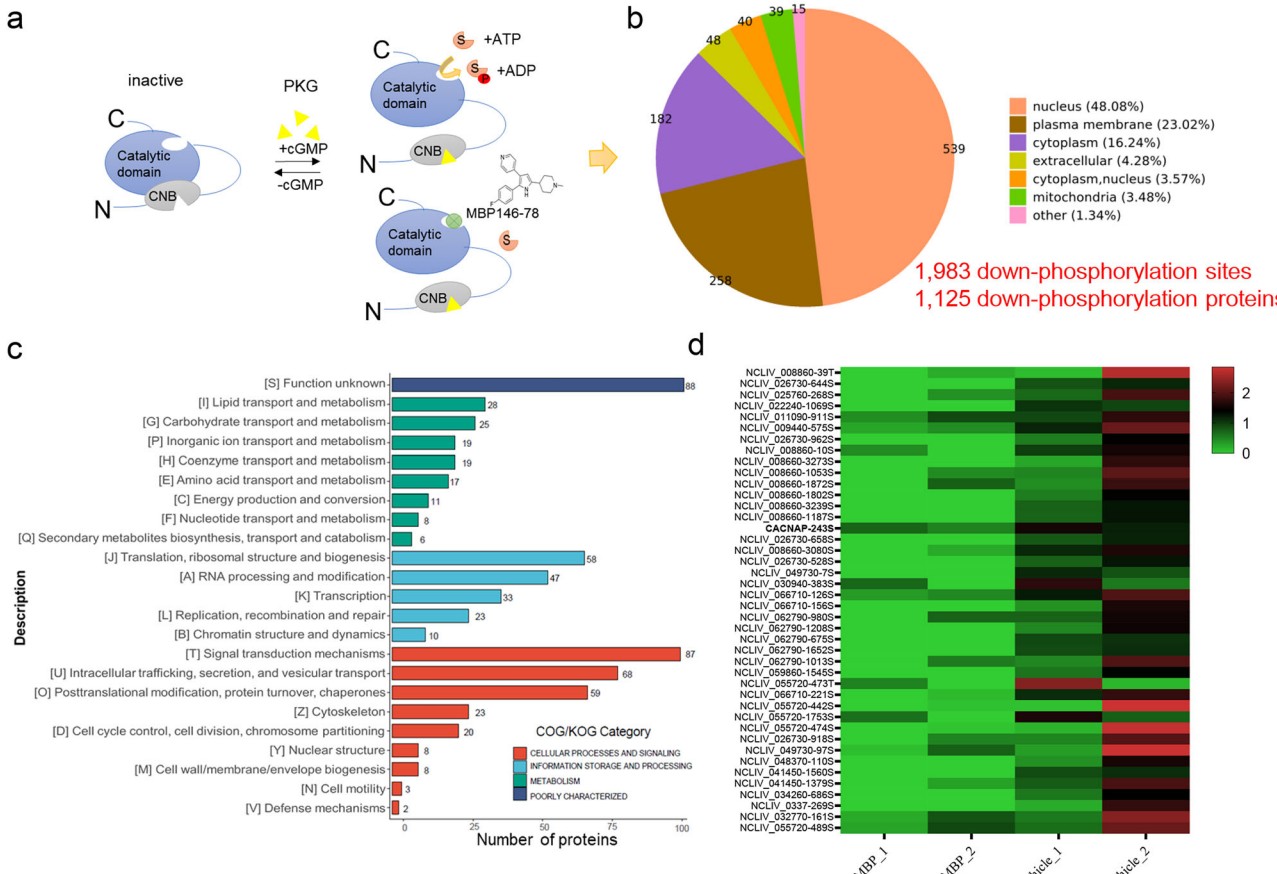

**Fig. 3 | Quantitative phosphoproteomics reveals PKG-regulated downstream molecules. a** Schematic diagram of PKG downstream protein screening by quantitative phosphoproteomics. **b** Predicted subcellular localization of 1125 potential PKG-regulated phosphorylated proteins. **c** KOG functional analysis of 1125 potential PKG-regulated phosphorylated proteins. **d** Ion transport and metabolism-related protein and phosphorylated modification site downstream of PKG.

cytosolic calcium ion concentration (Supplementary Fig. 1c, d) as described before[29,52]. The parasite fluorescence was measured over time series after stimulation with 8-Br-cGMP (5 μM) in both IAA treated and untreated *pkg*-mAID/GCaMP6f strains. Simultaneously, the IAA treated Nc1/GCaMP6f strain served as a control to account for any potential effects of IAA on [Ca$^{2+}$] levels. The results demonstrated a transient and significant increase in fluorescence intensity in both the *pkg*-mAID/GCaMP6f and Nc1/GCaMP6f strains upon the addition of 8-Br-cGMP (Fig. 2k), indicating a transient increase in cytosolic calcium concentration. However, when PKG was degraded by IAA treatment, the fluorescence response to 8-Br-cGMP was not significantly altered (Fig. 2k), suggesting that PKG activation is crucial for inducing calcium fluctuations in *N. caninum*. Collectively, our results underscore the multifaceted impact of PKG depletion on the lytic cycle of *N. caninum* tachyzoites.

### Quantitative phosphoproteome analysis reveals that NcPKG affects various signaling pathways in *N. caninum*

As a protein kinase, PKG transmits signal by phosphorylating specific downstream proteins[25]. Analyzing the downstream proteins of NcPKG is essential to understand its role in invasion and egress mechanisms. Previous studies have demonstrated that MBP164-78 (compound 1) (Supplementary Fig. 2a) is a potent and selective cGMP-dependent protein kinase inhibitor, active in both *Plasmodium* and *T. gondii*. Similarly, *N. caninum* tachyzoites hardly formed plaques incubation with MBP146-78 (Supplementary Fig. 2b), indicating the sensitivity of *N. caninum* PKG to this inhibitor. The PKG inhibitor significantly impaired invasion, egress, motility, and microneme secretion, with no impact on intracellular proliferation (Supplementary Fig. 2c–i), consistent with phenotypes from PKG conditional degradation (Fig. 2a–j). To determine the direct effects of inhibitors on PKG

in *N. caninum*, the recombinant protein of PKG (rPKG) was obtained via an *Escherichia coli* expression system (Supplementary Fig. 2j) and analyzed for inhibitory activity. Result displayed that MBP164-78 significantly inhibited rPKG kinase activity in vitro (Supplementary Fig. 2k). Collectively, these experiments confirmed the direct inhibitory effect of MBP146-78 on *N. caninum* PKG.

*N. caninum* tachyzoites were incubated with the PKG inhibitor MBP164-78 to identify downstream proteins regulated by PKG. After inhibition, NcPKG lost its kinase activity, preventing phosphorylation of its downstream proteins. Quantitative proteomics was used to measure overall phosphorylation level of tachyzoites proteins, and differentially phosphorylated proteins were compared to the uninhibited controls (Fig. 3a). In total, 17,519 phosphorylated peptides and 3294 phosphorylated proteins were identified in *N. caninum* (Supplementary Fig. 3a). Of these, 1125 proteins (1983 phosphorylation sites) exhibited decreased phosphorylation after inhibitor treatment (Supplementary Table 1), potentially reflecting the downstream molecules regulated by PKG.

GO analysis showed that these phosphorylated proteins predominantly participated in crucial biological processes, including cell processes, metabolic processes, and physiological regulation. Their molecular functions encompassed binding, catalytic activity, molecular regulation, transcription regulation, and other essential functions (Supplementary Fig. 3b). KOG classification revealed that the potential downstream phosphorylated proteins of PKG are involved in important processes, including cell signal transduction, protein post-translational modification, vesicular transport, gene transcription and translation, lipid transport and metabolism, inorganic ion transport and metabolism, and more (Fig. 3c). These results suggest that PKG, as a signaling center, plays an important role in numerous core processes of *N. caninum*.

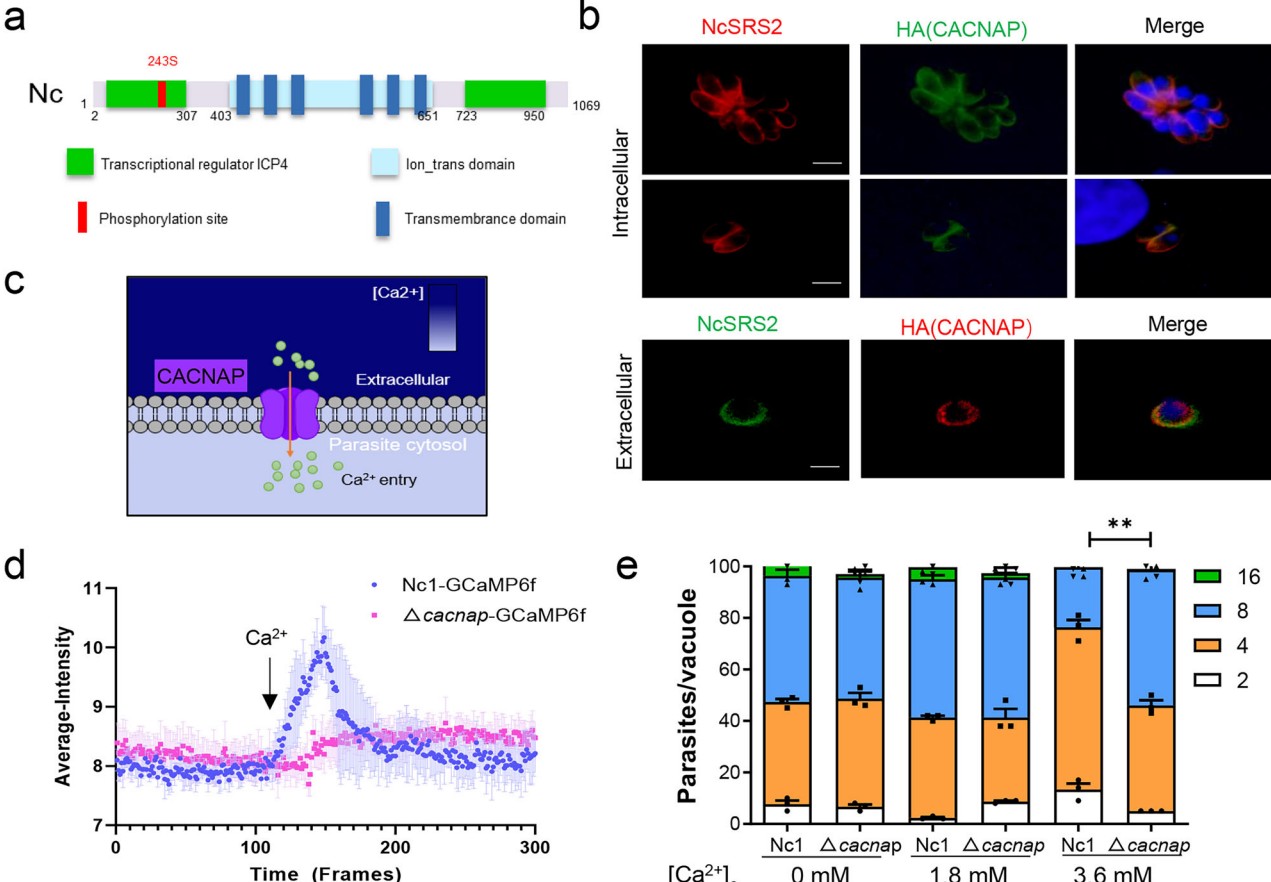

**Fig. 4 | CACNAP is associated with calcium channel proteins that affect calcium influx at the *N. caninum* plasma membrane. a** Schematic of CACNAP protein domains in *N. caninum*. **b** IFA determining of CACNAP subcellular localization in intracellular and extracellular parasites. CACNAP was stained with α-HA (green for intracellular, red for extracellular), plasma membrane with α-NcSRS2 (red for intracellular, green for extracellular), and nuclei with Hoechst (blue). Scale bars: 5 μm. **c** Functional prediction of CACNAP at the plasma membrane. **d** Continuous observation of calcium fluorescence signal between Δ*cacnap* and parental Nc1/GCaMP6f strains activated by 1.8 mM $[Ca^{2+}]_e$. Time-series images were converted to video and analyzed in ImageJ software. Fluorescence intensity in the GFP channel was quantified for 30 parasite regions at each time point and plotted over time using GraphPad Prism. **e** Comparison of Nc1 and Δ*cacnap* proliferation at different calcium ion concentrations. Data show means ± S.E. from 3 independent assays, \*\**p* < 0.01.

Considering the biological function of PKG in *N. caninum*, genes involved in parasite invasion, egress, and especially those regulated by $[Ca^{2+}]_i$ are of special interest. Among 19 proteins associated with inorganic ion transport and metabolism (Fig. 3d), NCLIV_005460 contains an ion_trans domain, described by the KOG database as a T-type voltage-gated $Ca^{2+}$ channel, pore-forming alpha1I subunit. This protein may be associated with $Ca^{2+}$ transport, a feature not been reported in other parasite, and its function warrants further characterization.

### NCLIV_005460 is associated with calcium channels proteins that affect calcium influx at the plasma membrane of *N. caninum*

NCLIV_005460 encodes 1069 amino acid residue (Supplementary Data 1) and possesses an ion transport domain (403–651 a.a.) and six transmembrane helices (Fig. 4a), characteristic of cation channel family transporters, suggesting a potential role in cation transport. Additionally, it contains two putative transcriptional regulatory regions (2–307 a.a., 723–950 a.a.). Homology analysis showed that the amino acid sequence of NCLIV_005460 is relatively conserved among apicomplexan parasites (Supplementary Fig. 4a), with the closest relation to TGME49_222060, the homologous protein of *T. gondii* (39.83% amino acid similarity), annotated as a cation channel family transporter. Interestingly, TGME49_222060 contains charged regions analogous to voltage sensor domains of calcium voltage-gated channels[53], indicating it potential as a calcium channel candidate.

Phyre2 protein structure prediction (Supplementary Fig. 4b) indicated that the Ion_trans domain of NCLIV_005460 exhibited the highest similarity (26%) to the voltage-gated calcium channel subunit alpha-1s (Template: c3jbrA) (Supplementary Fig. 4c). The structural similarity implies a functional resemblance, particularly in $Ca^{2+}$ transport.

Protein localization is essential for functional analysis. To determine the localization of NCLIV_005460, we fused the C terminus of the gene with a 3 × HA tag in the parental Nc1 strain using CRISPR/Cas9 technology (Supplementary Fig. 4d). Western blotting verified that the protein was expressed as predicted, with an approximate size of 120 kDa in the HA-labeled strain (Supplementary Fig. 4e). Immunofluorescence analysis revealed that NCLIV_005460 localizes to the parasite's plasma membrane in both extracellular and intracellular stage (Fig. 4b). This suggests that this protein may play a role in $Ca^{2+}$ transport at the plasma membrane (Fig. 4c).

Studies in *Toxoplasma* have shown that $Ca^{2+}$ influx occurs when tachyzoites are exposed to 1.8 mM extracellular $Ca^{2+}$ ($[Ca^{2+}]_e$)[32,54]. The influx of $Ca^{2+}$ from the external environment activates $[Ca^{2+}]_i$, enhancing parasite vitality, motility, and microneme secretion[32,33]. To test whether NCLIV_005460 functions in $Ca^{2+}$ transport at the plasma membrane (Fig. 4c), we completely knocked out it in the Nc1/GCaMP6f strain (Supplementary Fig. 5a, b). The parasites were exposed to 1.8 mM extracellular $[Ca^{2+}]_e$, and changes in cytoplasmic $[Ca^{2+}]_i$ flow were observed. The results showed that the fluorescence signal in the parental Nc1/GCaMP6f strain significantly enhanced with 1.8 mM $[Ca^{2+}]_e$ in the extracellular buffer (Fig. 4d, blue), indicating an increase in $[Ca^{2+}]_i$, while the NCLIV_005460 knocked out strain showed minimal response (Fig. 4d, pink). Taken together, these data reveal that NCLIV_005460 is involved in extracellular

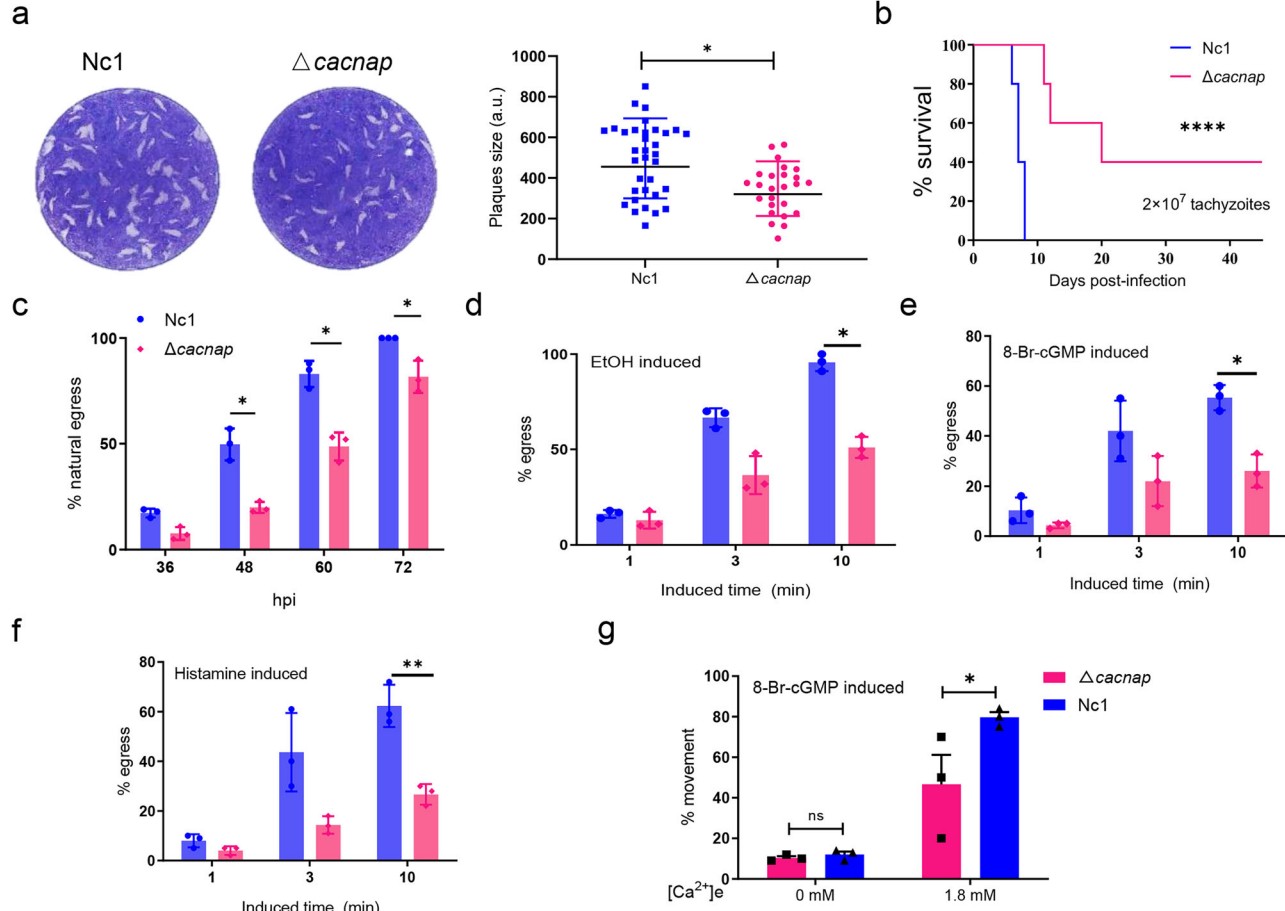

**Fig. 5 | Δcacnap parasites display defects in egress and motility. a** Comparison of plaque formation ability to analyze the growth of Δ*cacnap* and Nc1 strains in vitro. GraphPad Prism 9.0 was used for data statistical analysis. Data show mean values with S.E. from 3 independent assays, ***$p < 0.001$. **b** Mouse virulence experiment: A lethal dose ($2 \times 10^7$) of Δ*cacnap* and Nc1 strains were injected intraperitoneally into BALB/c mice ($n = 5$), and the survival was observed over 45 days. Statistical differences in mouse survival curves were analyzed using Curve comparison.

****$p < 0.0001$. **c** Statistics of natural egress proportions in Δ*cacnap* and Nc1 strains at different time points. Proportion of egressed parasites in Δ*cacnap* and Nc1 strains under different induction conditions: **d** EtOH, **e** 8-Br-cGMP, and **f** histamine. **g** Comparison of parasite motility in Δ*cacnap* and Nc1 strains under varying calcium ion concentrations. Data show mean values with S.E. from 3 independent assays, *$p < 0.05$, **$p < 0.01$.

calcium influx. Consequently, NCLIV_005460 was designated as *N. caninum* Ca²⁺ channel-associated protein (NcCACNAP).

To validate these findings, the parental Nc1 and Δ*cacnap* strains were incubated with different [Ca²⁺]ₑ concentrations and subsequently cultured in host cells to observe their proliferation. High concentrations of [Ca²⁺]ₑ pose potential dangers to cells[55,56], as they may precipitate inorganic and organic anions, leading to an imbalance in phosphate energy metabolism. The results demonstrated that there was no significant difference in proliferation between Nc1 and Δ*cacnap* in the absence of [Ca²⁺]ₑ (0 mM) or at physiological concentration (1.8 mM) (Fig. 4e). However, when exposed to a high [Ca²⁺]ₑ (3.6 mM) environment, the proliferation ability of Nc1 was significantly impaired, while Δ*cacnap* maintained normal proliferation (Fig. 4e). These findings suggest that the loss of CACNAP prevents damaged growth induced by high [Ca²⁺]ₑ, highlighting the role of CACNAP in modulating Ca²⁺ influx at the plasma membrane.

### Δcacnap parasites exhibit defects in egress efficiency and motility

Functional analysis revealed a significant reduction in plaque formation area following CACNAP deletion (Fig. 5a, Supplementary Fig. 5f, g). Virulence tests in mice further demonstrated that Δ*cacnap* strain exhibited significantly reduced virulence compared to the parental strain, although mice infected with Δ*cacnap* still lethal (Fig. 5b). These findings indicated that while CACNAP is important, it is not essential for the growth of *N. caninum*.

As phosphoproteomics identified CACNAP phosphorylation was down-regulated after PKG inhibition, we investigated whether CACNAP deletion phenotype aligns with PKG function. Compared to Nc1, the Δ*cacnap* strain showed no significant difference in intracellular proliferation, invasion, or microneme secretion (Supplementary Fig. 5c–e). However, its natural egress ability was significantly decreased (Fig. 5c). Further analysis of egress phenotype under various stimuli revealed that Δ*cacnap* parasites had significantly lower egress rates than that the parental strain. Ethanol (EtOH), which induces Ca²⁺ release from intracellular stores by activating phospholipase C[57,58], 8-Br-cGMP, which triggers Ca²⁺ release via PKG activation, and histamine, which activates Ca²⁺ release in the host cell[32], were used as induction conditions (Fig. 5d, f). These results indicate that CACNAP plays a crucial role in the egress process of *N. caninum*. Additionally, CACNAP deficiency impaired parasite motility (Fig. 5g), with restricted motility likely being the main reason of reduced egress ability.

Unexpectedly, the complemented strain (Δ*cacnap*/*CACNAP*) or phosphorylated modification site S243A and S243D mutant strains displayed significantly reduced plaque formation compared to the parental strain Nc1 and Δ*cacnap* (Supplementary Fig. 6c, d). Considering that CACNAP expression was maintained at a low level in *N. caninum*, qPCR was employed to examine CACNAP mRNA transcript levels in various regulated strains to investigate this phenomenon. CACNAP expression in the complementation and phosphorylation site mutants was over ten times higher than in wide-type strains (Supplementary Fig. 6e). The

overexpression is likely due to the strong tubulin promoter used in the complement or mutant strains. A high concentration of calcium ions had shown to impair parasite growth (Fig. 4e), we speculated that the overexpression of CACNAP may enhance extracellular calcium ion influx, causing "calcium overload" and thereby affecting tachyzoite growth. In addition, the mRNA level of NCLIV_020420, a putative voltage-dependent L-type calcium channel subunit, was increased in CACNAP-deficient strains (Supplementary Fig. 6f), suggesting that CACNAP may have similar functions to calcium channels, and the parasite may rescue the impaired function for CACNAP loss by upregulating the expression of NCLIV_020420 to maintain the survival of *N. caninum*.

Collectively, PKG is essential for tachyzoite invasion and egress from host cells, as it promotes the secretion of microneme and enhances tachyzoite gliding motility by activating intracellular $Ca^{2+}$ levels ($[Ca^{2+}]_i$). CACNAP participates in calcium influx at the plasma membrane, and its protein phosphorylation is down-regulated after PKG inhibition as shown by phosphoproteome. Loss of CACNAP results in delayed egress and impaired gliding motility, whereas high CACNAP expression negatively affects parasite growth. Not satisfactory, more work needs to be done to demonstrate the connection between PKG and CACNAP.

## Discussion

The key molecules involved in cyclic nucleotide signaling represent potential therapeutic targets for apicomplexan parasite diseases[35,59,60]. cGMP, synthesized by guanylate cyclases (GCs)[61], is involved in a variety of physiological processes. As a feedback mechanism, cGMP is hydrolyzed by cyclic nucleotide phosphodiesterases[62], which together regulate the levels of this intracellular messenger that mediates various functions in eukaryotes[16,63]. P4-ATPases generate membrane bilayer lipid asymmetry by translocating phospholipids from the outer to the inner leaflet[64–66]. GCα has been shown to be a critical regulator of PKG and that its associated P4-ATPase domain plays a primary role in generating cGMP for *T. gondii* tachyzoites[67] or malaria merozoite egress[16]. PKG serves as the primary intracellular receptor for cGMP, mediating various downstream physiological functions. However, the downstream events of PKG remain largely unclear.

In certain tissue cysts-forming coccidia, including *Toxoplasma*[42] and *Eimerella*[68], two isoforms of PKG are expressed by alternative translation initiation. The larger translated protein, such as TgPKG-I, localizes to the plasma membrane via N-acylation, governing PKG function[22]. The slightly smaller protein, TgPKG-II, remains in the cytosol due to the absence of N-terminal acylation residues[22]. The nucleotide sequence of *N. caninum* PKG closely resembles that of *T. gondii*. Similarly, *N. caninum* PKG expresses two isoforms of proteins, which localize to the plasma membrane and cytoplasm (Fig. 1c, d). This broad localization may correlate with the complexity of downstream regulated molecules.

In this study, the mAID system was employed to conditionally regulate PKG in *N. caninum*. First, the codon-optimized auxin receptor TIR1 was expressed in the cytoplasm of the Nc1 strain. Second, the C-terminal region of NcPKG was fused with the degradation elements mAID and Ty tags using CRISPR/Cas9 technology (Fig. 1b). Addition of auxin activated the SCF-TIR1 complex, leading to ubiquitylation of the mAID-tagged protein and eventually targeting it to the proteasome for degradation[22,43,69]. Using this genetic approach, we found that PKG contributes to invasion, egress, and motility. Additionally, we confirmed that PKG activation is essential for microneme secretion and calcium mobilization in *N. caninum* (Fig. 2).

Most of the insights into *T. gondii* egress are derived from directly or indirectly increasing the cytosolic $Ca^{2+}$ concentration to induce parasite egress from host cells[70]. The elevated $Ca^{2+}$ levels can stimulate the secretion of microneme and facilitate gliding motility, thereby promoting parasite egress[31]. Several studies have identified signal transduction proteins upstream of $Ca^{2+}$ release. In brief, guanylate cyclase (GC) senses signal molecules and produce cGMP, thereby activates PKG[47], which positively regulates various downstream components. These include the phosphatidylinositol metabolic pathway that triggers PI-PLC activation, which

generates the second messengers diacylglycerol (DAG) and $IP_3$ and ultimately results in microneme secretion[8]. $IP_3$ is believed to promote calcium release from intracellular stores. Consequently, calcium-dependent protein kinases (CDPKs) are activated to regulate microneme secretion and motor system activation. DAG is phosphorylated by diacylglycerol kinase, generating PA, sensed by an acylated pleckstrin-homology (PH) domain-containing protein (APH) on the microneme surface docking for microneme protein release[71]. Additionally, crosstalk between PKA and PKG controls pH-dependent host cell egress of *T. gondii*, while PKAc1 is presumed to regulate at least one cGMP phosphodiesterase PDE2 for limiting cGMP production and negatively modulating calcium ion signaling during the egress process[72–76]. As a signal transduction center, PKG senses upstream cGMP second messenger signal and forms signal cascades through the phosphorylation of specific downstream proteins to exert different physiological functions[77,78]. To better understand how NcPKG regulates invasion and egress in *N. caninum*, it is necessary to screen for and identify its downstream proteins. However, the transient occurrence of protein phosphorylation and the low abundance of phosphorylated proteins in vivo create certain difficulties in identifying protein kinase substrates. Both qualitative and quantitative phosphorylationomics have been applied in apicomplexan parasite to analyze molecular differences in different cycles or strains[25,79,80]. In this study, we employed chemical inhibition combined with quantitative phosphoproteomics to reveal the phosphorylation events mediated by NcPKG. Generally, sites or proteins with decreased phosphorylation after PKG inhibition are considered PKG-regulated, and we will focus on these proteins. Analysis revealed that compared to the wild type group, many sites could not be quantified following inhibitor treatment. This may be due to the phosphorylation level of PKG being below the detection threshold or due to the inhibition of PKG preventing phosphorylation. These sites or proteins are more likely regulated by PKG. Therefore, the statistics of these proteins, along with the significantly different proteins analyzed in aforementioned phosphoproteomics, as classified as potential downstream phosphorylation sites or proteins. In total, there are 1983 phosphorylation sites and 1125 corresponding phosphorylated proteins (Fig. 3b). These potential downstream molecules of NcPKG are involved in cell signaling, lipid transport and metabolism, ion transport, transcriptional regulation, and vesicle transport, indicating the broad role of PKG during the tachyzoite stage of *N. caninum*. However, PKG inhibition may decrease the activity of some downstream protein kinases[25], meaning that not all selected sites or proteins with reduced phosphorylation levels are directly regulated by PKG.

Before egress, during gliding movement, or invasion, the cytoplasmic $Ca^{2+}$ level substantially increases, primarily due to the release of intracellular calcium stores, such as the endoplasmic reticulum (ER), the plant-like vacuole [or vacuolar compartment] in *T. gondii*, and the food vacuole in *Plasmodium* spp[81]. Additionally, calcium influx from the extracellular environment enhances the motility and invasion of extracellular parasites[54] and rapidly increases the plasma calcium concentration for intracellular parasites, subsequently accelerating egress from host cells[33]. Elevated $Ca^{2+}$ activates a range of effectors, including a group of "plant-like" CDPKs, calcium-binding proteins, and proteins involved in vesicle exocytosis, to regulate various biological processes[82,83]. Numerous chemogenetic and genetic studies have demonstrated that PKG acts upstream of $Ca^{2+}$ signal generation and the $Ca^{2+}$-dependent pathways, thereby controlling key life cycle stages of apicomplexan parasites. However, the mechanisms by which PKG triggers calcium signaling and the molecules involved in calcium mobilization remain challenging to identify. An important study in *Plasmodium* revealed that the phosphatidylinositol metabolic pathway links PKG to calcium signaling[27]. Inhibition of PKG impedes the hydrolysis of phosphatidylinositol (4,5)-diphosphate, resulting in reduced production of inositol triphosphate ($IP_3$)[32], and subsequently decreasing the release of calcium ions from ER calcium stores. However, the regulatory mechanism by which PKG regulates $Ca^{2+}$ release remains unclear due to the absence of a receptor for $IP_3$ in apicomplexan parasites.

The screening of proteins downstream of NcPKG identified several molecules associated with phosphatidylinositol metabolism, such as phospholipase, patatin family protein (NCLIV_001050), putative PIK3R4 kinase-related protein (NCLIV_054290), and putative phosphatidylinositol 3-kinase (NCLIV_052580), etc. These proteins are likely involved in the activation of calcium signaling in parasites. Similarly, key molecules in downstream calcium signaling pathways were identified, such as calmodulin-like domain protein kinase isoenzyme gamma, related (NCLIV_011980), a member of the CDPK family, and p25-alpha family protein, related (NCLIV_030630), which has a calcium-binding domain and may play a key role in the invasion and egress of *N. caninum*. These findings confirm the close association between PKG, phosphatidylinositol metabolism, and the calcium signaling pathway.

A recent study identified ICM1, a PKG-interacting multi-channel membrane protein, in *P. falciparum* that shares homology with both transporters and calcium channels, suggesting its potential involvement in mobilizing of calcium from intracellular stores[34]. Orthologs of *Plasmodium* ICM1 are dispensable for $Ca^{2+}$ mobilization in *Toxoplasma gondii*[84]. Moreover, PKG has been reported to phosphorylate the Cav1.2 alpha1c and beta2 subunits, thereby modulating the function of the voltage-dependent $Ca^{2+}$ channel [Ca(v)1.2] in HEK cells[77]. Several pharmacological and genetic studies have demonstrated that intracellular signaling pathways involving cGMP, PKG, $Ca^{2+}$, and phosphatidylinositol phospholipase C regulate $Ca^{2+}$ influx, thereby triggering the lytic cycle characteristic of apicomplexan parasites[85].

Therefore, our focus shifted to proteins involved in ion transport and metabolic processes. NCLIV_005460 (CACNAP), featuring an ion transport domain and its homologs remain uncharacterized in the apicomplexan parasites. CACNAP localized at the plasma membrane of parasite, deletion of the gene mutants resulted in an inability to respond to extracellular calcium stimulation (Fig. 4d), indicating that CACNAP was involved in the influx of calcium from the plasma membrane (Fig. 4c). Extracellular $Ca^{2+}$ is thought to accelerate the lytic cycle[33]. The compromised egress and motility observed in parasites lacking CACNAP may stem from inhibited $Ca^{2+}$ activation caused by reduced extracellular $Ca^{2+}$ influx. There are still many questions to be studied.

The elevated expression of CACNAP can be attributed to the strong activity of the tubulin promoter used for ectopic complementation. However, this heightened expression resulted in unintended damage to the parasite, consistent with the detrimental effects of "calcium overload" on tachyzoites. Additionally, efforts were undertaken to construct a complement strain using the endogenous CACNAP promoter. Nevertheless, the low expression level of CACNAP in *N. caninum* made it challenging to monitor protein levels and precluded the stable establishment of an in situ complemented strain suitable for analysis. Consequently, the experiment designed to evaluate the phenotype associated with CACNAP complementation was ultimately discontinued. Nonetheless, distinct monoclonal knockout strains targeting CACNAP produced compelling results, highlighting the role of CACNAP in modulating egress progression in *N. caninum*. Phosphorylation of CACNAP is significantly downregulated upon PKG inhibition in *N. caninum*. However, the relationship between PKG and CACNAP with regard to the observed phenotypes remains inconclusive, and further experiments are required to ascertain the importance of S243 phosphorylation.

Although the function of this protein during the process of calcium influx has been evidently shown, CACNAP channel may also be responsible for the transport of $Na^+$ or $K^+$ (influx or efflux), as it is predicted to be a member of the cation channel transporter family. It is also known that there are several transporters using different ligands than transported ions, such as $K^+$-dependent $Na^+/Ca^{2+}$-exchangers[86] or $Ca^{2+}$ activated $K^+$ channels[87]. Additional assays are needed to test the functionality of CACNAP channel on sodium and potassium fluctuation. Since the role of both potassium and calcium has already been demonstrated on the successful egress of parasite *T. gondii*[33], it would be of interest to determine whether *N. caninum* follows a similar mechanism for host cell egress.

Collectively, we have substantiated that PKG plays a crucial role in regulating invasion, egress, motility, microneme secretion, and mobilization of $[Ca^{2+}]_i$ in *N. caninum*. CACNAP, participates in calcium influx at the plasma membrane, influencing egress and motility. Future research will aim to precisely identify PKG downstream proteins and uncover novel proteins associated with invasion or egress.

## Materials and methods
### Chemicals and antibodies
The chemicals utilized in this study, including 3-indoleacetic acid (IAA/auxin), pyrimethamine, fluorodeoxyribose (FUDR) drugs, chloramphenicol drugs, ampicillin, 8-Br-cGMP, and A23187, were purchased from Sigma-Aldrich unless specified otherwise. Dimethyl sulfoxide (DMSO) was purchased from Thermo Fisher Scientific. The PKG inhibitor MBP146-78 was sourced from MedChemExpress (MCE). The Kinase-Lumi™ chemiluminescence Kinase detection kit, was obtained from Beyotime Biotechnology, Shanghai. High purity, endotoxin-free plasmid extraction kits were purchased from Aidlab Biotechnologies, Beijing. The multi-fragments ligation kit and TransStart® Tip Green qPCR SuperMix were acquired from TransGen Biotech, Beijing. Commercial antibodies, including mouse anti-hemagglutinin (HA) (Cat no: H3663), rabbit anti-hemagglutinin (HA) (Cat no: SAB5600116), mouse anti-Flag (Cat no: B3111) and rabbit anti-Flag (Cat no: SAB4301135), were obtained from Sigma-Aldrich. Second antibody, Fluorescein (FITC)–conjugated Goat Anti-Mouse IgG(H + L) (Cat no: SA00003-1), Fluorescein (FITC)–conjugated Goat Anti-Rabbit IgG(H + L) (Cat no: SA00003-2), Cy3–conjugated Goat Anti-Mouse IgG(H + L) (Cat no: SA00009-1), Cy3–conjugated Goat Anti-Rabbit IgG(H + L) (Cat no: SA00009-2) were purchased from Proteintech, Wuhan, China. HRP–conjugated Goat Anti-Mouse IgG(H + L) (Cat no: IS001), HRP–conjugated Goat Anti-Rabbit IgG(H + L) (Cat no: IS003) were purchased from M&C Gene, Beijing, China. Additionally, antibodies like mouse anti-Ty[88], mouse anti-NcMIC4[89], mouse anti-NcMIC6[89], mouse anti-NcMIC11[90], mouse anti-actin[91], and rabbit anti-NcSRS2[91], were previously generated and preserved in China Agricultural University, Beijing for use in this study.

### Parasites and host cells culture
Tubulin promoter-driven OsTIR1 sequence (Supplementary Data 1) was inserted into the non-coding region between NCLIV-058880 and NCLIV-058890 genes using the CRISPR-Cas9 system to generate NcTIR1 parasite strain. *N. caninum* tachyzoites, including the aforementioned Nc1 and NcTIR1 strains, were cultured in human foreskin fibroblasts (HFF, ATCC, SCRC-1041) using Dulbecco's Modified Eagle Medium (DMEM) (Invitrogen, California, United States) supplemented with 10% fetal bovine serum (D10), penicillin (100 U/mL), and streptomycin (100 μg/mL). The cultures were maintained at 37 °C with 5% $CO_2$. Tachyzoites were inoculated when host cell confluency reached 80% or higher. Free tachyzoites were directly inoculated into the culture medium of new host cells. Approximately 1 h post invasion, the medium was replaced with DMEM containing 2% fetal bovine serum (D2). When the parasites began egressing (~48–72 h post-infection), a small number were transferred to new cells for further passage.

### Plasmid construction
The primers utilized in this study are detailed in Supplementary Data 3. Synthetic DNA fragments and primers were procured from Sangon Biotech (Shanghai) Co., Ltd. DNA fragments were amplified by PCR using Phanta Max Super-Fidelity DNA Polymerase (Vazyme Biotech, Nanjing, China) and ligated using seamless cloning kits (Vazyme Biotech, Nanjing, China) following the manufacturer's instructions. Plasmid sequences were verified by Ruibiotech Biotech (Beijing) Co., Ltd.

The ToxoDB Sequence Retrieval tool was used to locate the NcPKG gene (NCLIV_055260) C-terminal region, specifically the 400 bp of DNA following the termination codon, to generate the pNc_Cas9-CRISPR::sgNcPKG plasmid. The EuPaGDT Library in ToxoDB, an online

gRNA design tool, was utilized to design a site-specific gRNA. Using the pNc_Cas9CRISPR::sgNcUPRT plasmid (previously cloned by Yang[92] and preserved at China Agricultural University, Beijing) as a template, the UPRT-specific gRNA was replaced with a gRNA targeting NcPKG. The NcU6 promoter fragment, AMP-resistant fragment, and Cas9 fragment were then amplified. The NcPKG-specific gRNA was inserted at the 5 'end of the primer downstream of the U6 promoter and the 5' end upstream of the Amp-resistance fragment. This plasmid induced DNA double-strand breaks at the target site, with Cas9 expression driven by the SAG1 promoter and sgRNA expression by the NcU6 promoter. Other CRISPR plasmids were constructed using the same procedure, with specific plasmids and primers listed in Supplementary Data 3.

To generate the pUC19-NcPKG-mAID-Ty-DHFR plasmid, designed for homologous recombination, two segments (Flanks) were created before and after the NcPKG stop codon. These segments were amplified from the genomic DNA of Nc1 parasites, enabling homologous recombination with the genome of *Neospora* after target cleavage. The plasmid psLinker-mAID-Ty-DHFR (previously cloned by Long[88] and preserved at China Agricultural University, Beijing) served as the template. The mAID-Ty-DHFR fragment comprising the degradation element mAID sequence, the endogenous tag Ty, and pyrimethamine-resistant DHFR gene, was amplified for plasmid construction.

To generate the pLIC-CACNAP-HA-DHFR plasmid, the 5′ and 3′ flank sequences of the CACNAP, covering regions before and after the stop codon, were amplified from the genomic DNA of Nc1 parasites. The plasmid pLIC-DHFR-*grx* S14-HA (originally cloned by Song[93] and preserved at China Agricultural University, Beijing) served as the template. The HA-DHFR fragment and pLIC frameworks were then amplified for plasmid construction.

To generate the pTCR-*cacnap*::DHFR plasmid, the 5′ and 3′flank sequences, spanning regions before and after the complete CACNAP sequence, were amplified from the genomic DNA of Nc1 parasites. The plasmid p5′*grx* S14-DHFR-3′*grx* S14 (originally cloned by Song[93] and preserved at China Agricultural University, Beijing) served as the template, with the DHFR fragment and frameworks amplified for plasmid construction.

To generate the pUC19-HXGPRT::cacnap/S243A/S243D-Flag-CAT plasmid, the 5′ and 3′flank sequences, spanning the regions before and after the complete HXGPRT sequence, were amplified from the genomic DNA of Nc1 parasites. The plasmid p5′UPRT-Tubulin promotor-*grx*1-HA-DHFR-3′UPRT (originally cloned by Song[93] and preserved at China Agricultural University, Beijing) served as the template, with the Tubulin promoter fragment and frameworks amplified. Simultaneously, the plasmid pLIC-FLAG-CAT-NcDcp1 (previously cloned by Wang[94] and preserved at China Agricultural University, Beijing) was used as the template for amplify the Flag tag and chloramphenicol resistance gene (CAT). The coding sequences of CACNAP or CACNAP^{S243a/s243d} were amplified using cDNA of Nc1 as a template for plasmids construction.

To generate other pUPRT-pTUB-GCaMP6f plasmid, the 5′ and 3′ flank sequences, spanning the regions before and after the complete UPRT sequence, were amplified from the genomic DNA of Nc1 parasites. The plasmid p5′UPRT-Tubulin promotor-*grx*1-HA-DHFR-3′UPRT (originally cloned by Song[93] and preserved at China Agricultural University, Beijing) served as the template for amplifying the Tubulin promoter fragment. The plasmid pTgUPRT_DHFR-pTUB-GCaMP6f (Provided by Silvia N.J. Moreno and preserved at China Agricultural University, Beijing) was served as the template to amplify the GCaMP6f fragment and frameworks for plasmid construction.

### Generation of transgenic parasites
CRISPR/Cas9 gene editing was used for the generation of transgenic parasites, following the procedures described previously[92]. For stable transformants, $1 \times 10^7$ freshly harvested parasites underwent transfection with a purified gRNA-specific CRISPR/Cas9 plasmid and linearized dsDNA through electroporation. Transgenic parasites were selected based on antibiotic resistance. For instance, *pkg*-mAID, *cacnap*-HA, and Δ*cacnap* strains

were selected using pyrimethamine drugs (3 μM), while the GCaMP6f strain was selected using 5-fluorodeoxyuridine drugs (20 μM). The complemented strain (Δ*cacnap*/CACNAP) and phosphorylation site mutants (Δ*cacnap*/CACNAP^{243A}, Δ*cacnap*/CACNAP^{243D}) underwent selection on chloramphenicol (20 μM). Parasite clones were isolated through limiting dilution[91].

### Western blotting
Western blotting was employed for the characterization and quantification of tachyzoite proteins. Tachyzoites were collected. purified, and lysed on ice for 30 min using 50 μL RIPA lysis buffer and 1 μL protease inhibitor (both from Huaxinbio, Beijing, China). Subsequently, 10 μL of $6 \times$ SDS protein loading buffer was added to the protein samples and boiled for 5 min. Aftercooling, the proteins were separated by SDS-PAGE and transferred to nitrocellulose membranes. Primary antibodies included mouse or rabbit anti-HA (1:5000), mouse or rabbit anti-Flag (1:5000), mouse anti-Ty (1:50), mouse anti-NcMIC4 (1:300), mouse anti-NcMIC6 (1:1000), mouse anti-NcMIC11 (1:300), mouse anti-actin (1:5000), and rabbit anti-NcSRS2 (1:500). Secondary antibodies included HRP-labeled sheep anti-rabbit IgG (1:10,000) and HRP-labeled sheep anti-mouse IgG (1:5000).

### Immunofluorescence microscopy
Immunofluorescence microscopy was performed as previously reported[91] to analyze protein subcellular localization. HFF cells were cultured on coverslips placed in a 12-well plate, and tachyzoites were inoculated as required for the experimental. The coverslips were fixed in 4% polyformaldehyde for 15 min at room temperature, followed by permeabilization with 0.25% Triton X-100/ PBS buffer for 15 min. Subsequently, the cells were blocked with a 3% BSA/ PBS blocking solution for 1 h at 37 °C. Staining was carried out using appropriate primary antibodies, such as mouse anti-Ty (1:20), mouse anti-HA (1:100), rabbit anti-NcMIC6 (1:100), and rabbit anti-NcSRS2 (1:500). Fluorescently labeled secondary antibodies specifically FITC/Cy3 labeled sheep anti-rabbit/mouse IgG (1:100) were used along with DAPI (5 mg/mL). Antibodies diluted in the blocking solution were incubated for 1 h at 37 °C. After sealing the coverslips with an anti-fluorescence quencher, observations were made using fluorescence microscopy.

### Plaque formation
After counting, 200 tachyzoites were inoculated onto a 12-well plate containing HFF cells. Depending on the experimental purpose, either 500 μM IAA or an equivalent volume of DMSO was added to the medium, or different transgenic strains were inoculated. The cells were incubated for 7 to 9 days at 37 °C in a 5% $CO_2$ incubator. After the formation of macroscopic plaques, the culture medium was discarded, and the cells were gently washed once with PBS. Subsequently, the cells were fixed in 4% paraformaldehyde for 30 min at room temperature. The paraformaldehyde was discarded, and the cells were washed once with PBS. Staining was performed using 1 mL of crystal violet solution for 2 h, followed by another wash with PBS. Finally, the cells were dried inverted. The sizes of the plaques were measured using Photoshop software. GraphPad Prism 9.0 was utilized to analyze data differences and generate visual representations.

### Tachyzoite growth and replication assays
In the tachyzoite growth and replication assays, $2 \times 10^5$ tachyzoites were inoculated onto cell slides (M&C Gene Technology, Beijing) covered with HFF cells and cultured at 37 °C for 1 h. Non-invaded parasites were washed off with PBS. The medium was then replaced with either IAA, DMSO or inoculated with different transgenic parasites for an additional 30 h. The medium was discarded, and the cells fixed in 4% paraformaldehyde for 20 min before IFA. The primary antibody used was a rabbit anti-SRS2 polyclonal antibody (1:500), and the secondary antibody was Cy3-labeled sheep anti-rabbit IgG (1:100). The parasites were observed under an inverted fluorescence microscope, and the number of PVs containing 1, 2, 4, 8, and 16 or more parasites was counted. For each slide, 100 PVs were counted to analyze the proportion of each parasite number. The

experiments were repeated three times for statistical analysis. GraphPad Prism 9.0 was used to analyze difference in the data and generate plots.

### Invasion assay

Tachyzoites were cultured in medium containing 500 μM IAA or DMSO for 8 h prior to collection, or different transgenic strains were collected simultaneously. After counting, $2 \times 10^7$ parasites were inoculated onto cell slides coated with HFF. Following incubation at 37 °C for 1 h, non-invading parasites were washed off with PBS and fixed in 4% paraformaldehyde for 20 min prior to IFA. The paraformaldehyde was removed, the slides were washed three times with PBS, and then blocked with 3%BSA/PBS for 1 h. Before permeabilization, rabbit SRS2 polyclonal antibody (1:500) was used as the primary antibody and incubated at 37 °C for 1 h, followed by three washes with PBS. A permeabilizing solution was added for 15 min, followed by the addition of a blocking solution for 1 h. The primary antibody was mouse anti-IMC1 polyclonal antibody (1:300). The secondary antibodies were FITC-labeled sheep anti-rabbit (1:100) and Cy3-labeled sheep anti-mouse (1:100), and nuclei were stained with Hoechst solution (1:100). The number of host nuclei (blue), non-invaded parasites (green) and all parasites (red) were counted. For each cell slide, five visual fields were analyzed (upper, lower, left, right, and middle). Quantification involved counting blue host cell nuclei, green non-invaded parasites, and red total parasites. The number of successfully invaded parasites was calculated by subtracting the number of red parasites from the number of green parasites. Invasion efficiency was calculated as the ratio of successfully invaded parasites to host cell nuclei. This counting process was repeated three times for each slide to ensure accuracy and reliability. GraphPad Prism 9.0 was used to analyze differences in the data and plots.

### Induced egress assay and natural egress assay

To induce egress, $5 \times 10^4$ fresh tachyzoites were harvested and inoculated onto cell slides coated with HFF cells. Depending on the experimental design, the tachyzoites were cultured in D2 medium for 24 h, followed by replacement with medium containing 500 μM IAA or DMSO for an additional 8 h. Alternatively, different transgenic strains were inoculated simultaneously and cultured for 30 h. To induce parasite egress from host cells, calcium ionophore A23187 (1 μM) or 8-Br-cGMP (5 μM) was added for 1–10 min in an incubator at 37 °C. The medium was then discarded, and rapid fixation with 4% paraformaldehyde was performed to terminate egress, followed by IFA.

For natural egress, $2 \times 10^4$ fresh tachyzoites were collected and inoculated onto cell slides coated with HFF cells. The tachyzoites were cultured for 30, 48, 72, and 86 h, respectively. The cells were fixed with 4% paraformaldehyde and observed by IFA. The primary antibody was rabbit anti-SRS2 polyclonal antibody (1:500), and the secondary antibodies were Cy3-labeled sheep anti-rabbit (1:100) and DAPI solution (1:100). One hundred PVs were randomly selected and observed under an inverted fluorescence microscope to determine the proportion of egressing PVs. Each slide was counted five times for accuracy. GraphPad Prism 9.0 was used to analyze differences in the data and generate plots.

### Motility assay

The slides were placed in a 12-well plate and treated with 1 mL of poly-D-lysine solution (50 μg/mL) from Beijing Solarbio Science & Technology Co.,Ltd. at room temperature for 1 h. After remmoving the solution, the slides were washed twice with distilled water and allowed to air dry. Tachyzoites were collected, purified, and resuspended in extracellular buffer[22] (EC buffer, 5 mM KCl, 142 mM NaCl, 1 mM MgCl₂, 1.8 mM CaCl₂, 5.6 mM d-glucose, 25 mM HEPES, pH 7.4). An appropriate amount of tachyzoites was added dropwise to the cell slide, left on ice for 5 min, and the non-adherent parasites were washed off with pre-cooled PBS. Subsequently, EC buffer containing 5 μM 8-Br-cGMP was added and incubated at 37 °C for 30 min to generate traces of gliding movement. The EC buffer was discarded, and the samples were immediately fixed with 4% paraformaldehyde to terminate gliding movement; the trajectories were then

observed using IFA. The primary antibody was rabbit anti-SRS2 polyclonal antibody (1:500), and the secondary antibody was Cy3-labeled sheep anti-rabbit (1:100). The movement states of 100 tachyzoites were observed, and the number and proportion of tachyzoites with movement trajectories were counted. Each slide was examined three times, resulting in a total of three slides. The movement distance of the 50 moving parasites was measured, and the length of the movement tracks was analyzed in relation to the body length of *N. caninum*. GraphPad Prism 9.0 was used to analyze the differences in the data and generate plots.

### Microneme secretion

The $2 \times 10^7$ tachyzoites were collected, purified, resuspended, and washed with EC buffer. The precipitates were resuspended in 100 μL of EC buffer, and then treated with either 1 μM A23187 or 5 μM 8-Br-cGMP for 10 min at 37 °C. The excreted secreted antigen (ESA) of *N. caninum* was conducted by centrifugation at $800 \times g$ for 10 min at 4 °C. The secreted content of micronemes in the supernatant was analyzed by western blotting.

### Calcium measurements assay

Purified tachyzoites of the Nc1/GCaMP6f strain were suspended in PBS. For observation, tachyzoites were added to a glass bottom dish for image acquisition using a fluorescence microscope. The parasites were allowed to settle to the bottom for 1 to 2 min. Fluorescence images were collected in alternating phases using microscopy, capturing 300 consecutive images at 1-s intervals over 5 min. The 30th frame was treated with 1 μM calcium ionophore A23187, 5 μM 8-Br-cGMP, or 1.8 mM CaCl₂ (from Beijing Solarbio Science & Technology Co., Ltd.). The time-series images were converted into a video and imported into ImageJ software. Thirty parasite regions were selected, and the fluorescence intensity in the GFP channel was quantified at each time point. GraphPad Prism was used to plot the fluorescence intensity of the calcium indicator strain as a function of time.

### Prokaryotic expression of PKG recombinant protein

The gene encoding NcPKG was amplified using the cDNA of Nc1 strain as a template and ligated into the pGEX6p-1 vector to construct the prokaryotic expression plasmid pGEX6p-1-PKG. The prokaryotic expression plasmid was transformed into Transetta expression-competent cells of *Escherichia coli*. The transformed bacteria solution was gradually propagated and then added to 500 mL of LB medium containing kanamycin (Sigma-Aldrich, USA). The mixture was incubated on a horizontal shaker at 37 °C until it reached the logarithmic growth phase (~4–6 h). 0.8 mM IPTG (Merck Millipore Novagen, Germany) was added to the bacterial solution to induce expression, and the bacteria were incubated overnight at 16 °C on a horizontal shaker. The induced bacteria were collected by centrifugation at 4 °C, sonicated in an ice bath, and the supernatant was purified using a Sepharose 4B column (Merck Millipore Novagen, Germany).

### Kinase activity inhibition assay

Kinase activity inhibition experiments were performed according to methods previously described in the relevant literature[95]. Briefly, 25 μL of assay buffer (comprising 20 μM ATP, 40 μM phosphorylated peptide GRTGRRNSI-NH2, 2% (v/v) DMSO, 20 μM cGMP, or 1 μM MBP146-78) was added to the system, followed by the addition of 25 μL of purified PKG recombinant protein diluted to 2 nM with assay buffer, and heat-inactivated rPKG protein was used as a negative control. The kinase reaction was initiated by the addition of PKG and incubated for 40 min at room temperature. The chosen concentration of ATP and reaction time ensured the conversion of 20–90% of ATP to ADP, with optimal conditions being 30–70% conversion of ATP to ADP. The quantification of reactive PKG kinase activity was achieved by measuring the remaining ATP in the solution after the kinase reaction, employing the Kinase-Lumi™ Luminescent Kinase Assay Kit. For this purpose, 50 μL of the Kinase-Lumi™ chemiluminescence Kinase detection reagent was added and mixed, with the reaction proceeding for 10 min at room temperature. Chemiluminescence detection was carried out using a multifunctional microplate reader, and the

amount of remaining ATP in the sample wells was calculated based on a standard curve. PKG kinase activity was expressed as the ATP conversion rate, defined as the initial amount of ATP added minus the remaining amount of ATP divided by the initial amount of ATP added.

## Quantitative phosphoproteome analysis

The PKG inhibitor MBP146-78 was applied to *N. caninum* tachyzoite, and phosphorylated differential proteins were screened using LabelFree quantitative phosphoproteomics. The experimental method was adapted from literature related to *Plasmodium*[25], and the specific steps are detailed below. Nc1 tachyzoites were inoculated into HFF cells, and cultured for 3 days before purification and collection. The parasites were washed once with PBS and divided into two aliquots, each containing no less than $2 \times 10^8$ tachyzoites. After centrifugation, the cells were resuspended in 200 μL of RPMI 1640 medium (M&C Gene Technology, Beijing). The inhibitor MBP146-78 was added to one sample to achieve a final concentration of 10 μM, while an equal volume of DMSO was added to the other sample as a control. The cells were incubated at 37 °C for 1 h. To prevent protein dephosphorylation, samples were centrifuged at $2000 \times g$ for 5 min at 4 °C to collect the precipitate after incubation. The precipitate was resuspended in 50 μL of 0.01% (w/v) saponin (M&C Gene Technology, Beijing) in TBS (Solarbio Science & Technology Co., Ltd. Beijing). Protease inhibitors (Solarbio Science & Technology Co.,Ltd, Beijing) and phosphatase inhibitors (MedChemExpress, China) were added to prevent protein degradation and dephosphorylation. After incubation at 4 °C for 20 min, the cells were centrifuged at $8000 \times g$ for 15 min, and the supernatant was collected. For soluble protein collection, the precipitate was resuspended in 50 μL of non-denaturing protein lysate (Beijing Solarbio Science & Technology Co., Ltd.) with protease inhibitors and phosphatase inhibitors and incubated on ice for 10 min. Samples were sonicated for 3 s, followed by an additional two rounds of sonication, and then centrifuged at $8000 \times g$ for 15 min to collect the supernatant. Protein concentration was determined using the BCA Protein Quantification Kit (M&C Gene Technology, Beijing), ensuring that the total protein amount was not less than 1 mg.

The phosphoproteomics process and data analysis for this study were technically supported by PTM-Biolab Hangzhou. The process involved trypsin digestion of protein samples, enrichment of phosphomodified peptides, and analysis using liquid chromatography-mass spectrometry. Secondary mass spectrometry data were retrieved using MaxQuant (v1.6.15.0). For protein identification, the database used was Blast *N. caninum* Liverpool-572307_ NcaninumLIV_ToxoDB-52_20210524.fasta (7115 sequences). GO annotation of the identified proteins was performed using eggNOG-mapper software (v2.0). For differentially phosphorylated protein screening, the Fold Change was determined by calculating the ratio of the mean quantitative values of all biological replicates in the PKG inhibitor treatment group (MBP) to those in the control group (Vehicle) for each modification site. The coefficient of variation (CV) for the relative quantitative value of each modification site in two comparison pairs of samples was calculated as the significance index. Based on the above differential analysis, modifications with CV ≤ 0.1 and expression level difference of MBP/Vehicle group less than 0.5 were considered significantly down-regulated. This threshold was used to identify significant changes in phosphorylation levels.

## Real-time PCR

The Nc1 and various gene-regulated strains were cultured on cells until they reached the same state (prior to egress). Tachyzoites cultured in T25 cells were collected and purified by placing them in an ice bath for 5 min before sample collection. The medium was removed through centrifugation, and the purified tachyzoites were washed twice with pre-cooled PBS. RNA extraction was carried out from the freshly collected parasites using a rapid RNA extraction kit (Invitrogen, California, United States). Subsequently, cDNA synthesis was performed using a reverse transcription kit (Invitrogen, California, United States). The obtained cDNA served as a template for real-time PCR reactions, which were conducted using the TransStart® Tip

Green qPCR SuperMix kit (TransGen Biotech, Beijing). This enabled the quantitative analysis of gene expression levels in the cultured parasites.

The relative quantification algorithm -ΔΔCt was employed to assess the expression differences of target genes in various gene-regulated strains. The ΔCt of each group was determined by subtracting the Ct value of the target gene (Ct target gene) from the Ct value of its internal reference (Ct reference gene, actin). The ΔΔCt was obtained by subtracting the ΔCt of the control group (parental Nc1 strain) from that of the experimental group (gene-regulated strain). The fold difference was calculated using the formula: Fold difference = $2^{-\Delta\Delta Ct}$. The experimental results were presented as mean ± standard error (SE). One-way ANOVA was conducted using GraphPad Prism 9.0, and pairwise comparison of values in each group were performed using the Dunnett test. Significance levels were denoted as follows: * for $P < 0.05$ (significant difference), and ** for $P < 0.01$ (very significant difference). These symbols were used to indicate the statistical significance of expression differences among the gene-regulated strains compared to the parental Nc1 strain.

## Animals infection experiments

In the in vivo *N. caninum* virulence assay, 5-week-old female BALB/c mice (Charles River, Beijing) were used. We have complied with all relevant ethical regulations for animal use. The mice were housed in specific pathogen-free conditions, residing in filter-top cages, and provided with sterile water and food. Daily monitoring was conducted to record the health status of the mice, and they were sacrificed at the endpoint of the experiment. All animal procedures and experiments performed in this study were approved by the Laboratory Animal Welfare and Animal Experimental Ethical Committee of China Agricultural University (Certificate No. CAU-AW31901202-2-1).

The mice were randomly grouped and the experiments commenced 1 week after the initiation of rearing. The parental strain and various gene-regulated strains, each containing $2 \times 10^7$ tachyzoites, were inoculated into the mice. Except for the parasites infected, other feeding conditions remained the same. There are five mice in each cage was one group. Clinical symptoms, time of death, survival rate, and other indicators were monitored to analyze the effect of the gene on the pathogenicity in the mice. Statistical analysis was performed using GraphPad Prism 9.0 software, and differences in mouse survival curves were assessed through Curve comparison, providing valuable insights into the influence of gene regulation on *N. caninum* pathogenicity in vivo.

## Statistics and reproducibility

Statistical analysis was conducted using GraphPad Prism 9.0 software. Data are presented as means ± SD, with error bars indicating SD values. Various statistical tests, including two-tailed Student's *t* test, one-way analysis of variance (ANOVA), and two-way ANOVA, were applied based on experimental requirements. Differences in mouse survival curves were analyzed using Curve comparison. The significance levels were denoted as follows: $p > 0.05$ indicates no significance (n.s.), $*p < 0.05$, $**p < 0.01$, and $***p < 0.001$. Representative data were repeated a minimum of two times, each with at least two independent biological samples and three technical replicates.

## Reporting summary

Further information on research design is available in the Nature Portfolio Reporting Summary linked to this article.

## Data availability

Data supporting the conclusions of this article are included within the manuscript and its supplementary files. Source data for all graphs and plots in the paper can be found in files Supplementary Data 2. The phosphoproteomics data are available via the ProteomeXchange Consortium (https://proteomecentral.proteomexchange.org) and the iProX database (https://www.iprox.cn/) with the dataset identifier PXD062765 and IPX0011618000. The datasets generated and analyzed during the current study are available from the corresponding authors upon reasonable request.

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

## Acknowledgements

We are grateful to Professor Silvia N. J. Moreno (University of Georgia, GA, USA) for providing plasmid pTgUPRT_DHFR-pTUB-GCaMP6f. We also thank Professor Shaojun Long (China Agricultural University, Beijing, China) who provided the plasmid psLinker-mAID-Ty-DHFR and mouse anti-Ty antibodies. This work was supported by the National Natural Science Foundation of China (32273029 and 31972700).

## Author contributions

X.W. and J.L. designed the study and wrote the manuscript. X.W. performed all the experiments, analyzed the data, and generated the figures. K.G. and Z.S. assisted in data curation and investigation involving dynamic observation of calcium ion. Z.Y. performed the structure prediction and bioinformatics analysis, as well as formal analysis and data visualization for this study. Z.Z. assisted in the expression and purification of recombinant proteins and data collection. S.Y., K.G., and N.Y. assist in plasmid construction and animal experiments. J.L., Q.L., and L.W. contributed by supervising the project and editing the manuscript. All authors involved in the discussion and provided feedback on the manuscript.

## Competing interests

The authors declare no competing interests.
