## [Transparent Peer Review file · Communications Biology]

Unveiling the Impact of cGMP-Dependent Protein Kinase of *Neospora caninum* on Calcium Fluxes and Egress Functions Through Quantitative Phosphoproteome Analysis

Corresponding Author: Professor Jing Liu

Version 0:

Reviewer comments:

Reviewer #1

(Remarks to the Author)

The manuscript "Unveiling the Impact of cGMP-Dependent Protein Kinase of *Neospora caninum* on Calcium Fluxes and Egress Functions Through Quantitative Phosphoproteome Analysis" by Wang et al. used a mini auxin-inducible degron system to knock down cyclic GMP-dependent protein kinase G in *Neospora caninum* and revealed its indispensable role in tachyzoite invasion and egress, highlighting 1125 potential downstream targets involved in critical cellular functions and identifying CACNAP as a potential key player in calcium influx.

Although PKG has been investigated in related parasites (e.g., *Toxoplasma*), this is still an interesting paper that provides significant insights into how PKG in *Neospora caninum* regulates invasion and egress. However, the claim that PKG functions through phosphorylation of CACNAP at Serine 243 was not backed by a lot of experimental data and needs further experimentation.

Main Comments:

- 1) The authors were not able to complement the CACNAP knockout phenotype (Figure 6). They speculate that this was because of overexpression from the tubulin promoter. However, they should demonstrate that the CACNAP knockout phenotypes were indeed due to CACNAP deletion, and not to an off-target effect, by complementing with a CACNAP construct using the endogenous promoter. They should also show the role of the S243 phosphorylation by complementing the knockout with an S243 mutant expressed from the endogenous promoter. Currently the connection between PKG and CACNAP with regards the phenotypes observed is speculative as it was not shown that the S243 phosphorylation is important.
- 2) The authors claim that the S243D mutations lead to an increase in transcript expression (Fig 6e). However, it seems very unlikely that this amino acid change could lead to transcript level changes. It is more likely that when they complemented the knockout strain multiple copies of the complementation construct were inserted into the parasite genome.

Minor comments:

There were multiple instances of sentences that were not very clear. For example:

- 1) Line 76 resistance in door-spread parasites. What are door-spread parasites?
- 2) Lines 156-157 Analyzing the effect of PKG on the natural egress process, the tachyzoite of the PKG-mAID strain was simultaneously inoculated with multiple cell climbs. It is unclear what the authors mean by this (multiple cell climbs?)
- 3) Line 188 genetically encoded Ca²⁺-indicating manner -> Ca²⁺-indicator
- 4) 205 Previous studies have d that -> have demonstrated that
- 5) The *Toxoplasma* version of NCLIV_005460 was endotagged and described in J Eukaryot Microbiol. 2018 Jul; 65(5): 709–721. doi: 10.1111/jeu.12626, which the authors could cite.
- 6) 344 Science we have previously shown that -> Since we have previously shown that

Reviewer #2

(Remarks to the Author)

This is an interesting piece of work that explores the role of the cGMP-dependent protein kinase (PKG) of *Neospora caninum* using reverse genetics and a specific inhibitor. Roles for NcPKG in microneme secretion, invasion, cell motility,

and egress, operating upstream of calcium have been demonstrated, results which concur well with studies in other apicomplexans. Phosphoproteomics identified CACNAP, a calcium channel-associated protein and functional analysis of this protein is also presented. The experimental work is very carefully carried out and conclusions are consistent with the data in general. However, in my opinion this is not the case with the final section of the results that deals with the potential role of a phosphosite of CACNAP in its transcriptional control. I have added further details below of my issues with this section of the manuscript.

Introduction,

Line 52, I would refer to the protein as the 'cGMP-dependent protein kinase' and then 'PKG' thereafter. It is also known as 'protein kinase G'. Not 'cGMP-dependent protein kinase G'.

Line 54, I presume this should be the 'γ-phosphate group'.

Line 58-60, should there be a reference for the cGKI information? Perhaps all is within reference 8.

Line 76, is 'door spread parasites' intended? If so, reference 11 is not an appropriate citation.

PMID: 33762339 could be cited in relation to the involvement of PKG and calcium in apicomplexan parasites.

Line 106, perhaps replace 'intracellular' with 'apicomplexan'.

Results

Line 119, is it homology (BLAST) searching rather than secondary structure analysis that revealed this?

Lines 120-21, is it clear from the sequence data that all four CAP_ED domains are likely to be functional cGMP-binding domains, or only three of the four domains as is the case in e.g. *P. falciparum* where key conserved amino acids are absent from site C?

Lines 122-24, should be reworded.

Lines 128-30, only later in the discussion the significance of these two bands with respect to coccidian dual N-terminal (dual) acylation of PKG. I think it should be also mentioned briefly here.

Line 157, is 'cell climbs' intended? Cell cycles perhaps?

Line 167, this should be 'phosphodiesterases' rather than 'phosphodiester acid'.

To my knowledge, the reason that 8-Br-cGMP is effective is that it is one of two cGMP analogues that can bind apicomplexan PKG effectively (rather than the rate of hydrolysis). This was previously shown by e.g. Robert Donald and colleagues (PMID: 11914085).

Line 188, could this be changed to e.g. 'a genetically-encoded calcium sensor'?

Line 205, presumably d= demonstrated.

Lines 210-11, consider changing: 'indicating the sensitivity of MBP146-78 to the inhibitory activity of *N. caninum*' to e.g. 'indicating the sensitivity of *N. caninum* PKG to MBP146-78'.

Figure 3f, these data would be much more accessible to readers if the proteins (where annotations are available) are actually named to the left (rather than having NCLIV numbers). This would also allow readers to quickly identify which row corresponds to the putative calcium channel if it is labelled NcCACNAP and other proteins of interest. The text in Figure 3e and Supplementary Figure 3b is too small and should be enlarged.

Figure 6

In the legend to Figure 6c and e, 'com' should be defined. I assume that it is complementation with the wild type sequence. Line 329, I assume this refers to complementation of the knockout. State that complementation with one of two different mutations were performed in parallel.

Line 343. I assume that 'normal strain' means wild type?

Line 344, change 'science' to since.

Figure 6e, the RNA expression levels are very low in Nc1 (I assume this is wild type) and zero in the knockout (as expected, also, note that the y-axis of Figure 6e has no units).

Lines 346-55, I am confused as to whether the authors are saying that 1) the elevated transcription levels may be an artifact of the strong tubulin promoter used in the complementation strategy (and the consequent calcium overload), or 2) that this phosphorylation site could be involved in transcriptional regulation of the gene (especially since it is within a motif predicted to be involved in transcriptional regulation). The position of the authors on these two possibilities needs to be clarified. Also some explanation of why the phosphomimetic (S243D) apparently leads to the highest levels of RNA expression.

The subtitle of this section I think needs re-writing to reflect the results of these experiments (see below)*. The current subtitle I believe is an inaccurate reflection of the findings. Particularly since the S243A mutant (which will not be phosphorylated) also has a very high level of RNA expression.

Whilst the findings are interesting, I don't think any conclusions can be drawn from these experiments regarding the role of

this phosphorylation site in protein function. Therefore I disagree with the statement on lines 366-7. Perhaps the use of an alternative promoter (more appropriate than tubulin) in the complementation system would be a good strategy going forward. *In my opinion it would be a good idea to totally omit this section of the results from the paper and perhaps only mention very briefly in the Discussion that this was attempted, but that the results were inconclusive, and why. I believe there is no mention of these findings in the Abstract.

How do the authors envisage this protein interacting with the transcriptional machinery given its location in the cell.

Discussion

Line 372, insert: ...only 'one' gene encoding....

Line 373, as mentioned earlier, no analysis is presented regarding whether the four potential cGMP-binding domains of NcPKG are likely to be functional. In other apicomplexans there are four adjacent extended sequence motifs, three of which are functional and a fourth one lacks some key residues required for cGMP binding. The statement here implies that all four in Nc look to be functional. This needs clarification and justification here and earlier.

The Discussion reiterates the results too extensively as evidenced by citing the figures throughout.

Line 412, the term 'normal group' needs to be changed.

Lines 475-477, as mentioned previously, I don't think the statement here is supported by the data.

Reviewer #3

(Remarks to the Author)

Claims and the Novelty of the Manuscript

In this manuscript, Wang et al. aimed to explore the function of cGMP dependent protein kinase (PKG) specifically in tachyzoites of *Neospora caninum* by utilizing a mini auxin-inducible degron (mAID) system for conditional degradation of the protein. They demonstrated significant role of NcPKG on regulating motility, invasion and egress of the parasite through microneme secretion and increased intracellular calcium levels measured by genetically encoded calcium sensor GCaMP6f. Subsequently, recombinant NcPKG protein has been produced in *Escherichia coli*, and the impact of MBP164-78 (compound 1) to block kinase activity of PKG has been confirmed by in vitro assays. Subsequently, they attempted to find downstream targets of NcPKG by quantitative phosphoproteome analysis. As a result, 1125 proteins were found to be differentially phosphorylated/ downregulated in MBP164-78 treated conditions when compared to the normal state parasites. Out of these proteins, 19 have been implicated in the involvement of inorganic ion transportation and metabolism. In the last stage of the manuscript, the authors focused on the characterization of a novel protein with gene ID of NCLIV_005460 which was described as a T-type voltage-gated Ca²⁺ channel in KOG database and has not been reported in other apicomplexan parasite so far.

The work performed by the authors is very standard in other apicomplexan parasite *Toxoplasma gondii*; and the outcomes of the phenotyping assays in PKG-mAID conditional knockout strain are expected as shown in the depleted cGMP conditions of *T. gondii* tachyzoites (PMID: 29030485, 28465425, 30742070, 31235476, 30449726, 30992368, 351073373, 3335684, 36382189). However, it can be considered as a first elaborative research paper related to cGMP signaling in *Neospora caninum* and thus keeps its novelty. Since *Toxoplasma* and *Neospora* are closely related coccidians, I suggest authors to include aforementioned cGMP-related recent publications in the introduction part and evaluate their findings by comparing phenotypes (i.e. outcomes of plaque assays as well as motility, invasion, egress and replication profiles of the *N. caninum* tachyzoites together with the microneme secretion) observed in *Toxoplasma* at discussion session. The same comment applies for the calcium influx (PMID: 26933036 and 26933037) and phosphoproteomic analysis of the MBP164-78 treated parasites (PMID: 29030485, 36265000). Mentioning such references will strengthen the manuscript and increase the reliability of the outcomes.

Significance:

This manuscript would be interesting for the researchers studying signaling pathways in protozoan parasites, especially apicomplexans. The authors put a considerable effort to conclude with a nice story by finding out a novel protein as a PKG target and characterizing it as a voltage-gated Ca²⁺ channel, which links cGMP cascade with calcium signaling to regulate motility dependent invasion and egress events. This discovery will definitely attract to the researchers working on the different protozoan models to further explore the potential of the protein proposed as the ion channel.

I believe that the manuscript will deserve to be published in *Communications Biology Journal* after supportive revision of the research as suggested below:

Major Comments:

1. It is not clear how auxin sensitive TIR1 stain was generated.

Lines 390-392

"The codon-optimized auxin receptor Tir1 was expressed in the cytoplasm of Nc1 strain (originally cloned by Yang, preserved at China, Agricultural University, Beijing)"

For the reproducibility of the work by the others, it is essential to mention which region has been targeted in the genome to express TIR1 receptor, or how it was expressed in the cytosol. If the strain was developed using the same strategy as recently published by Mineo et al. (PMID: 35019667), then this paper should be cited properly.

2. In Fig 1f, Fig 5a and Fig 6d the analysis of the plaque area must be performed again using single plaque images and processing with the size measurements using ImageJ, as Photoshop unable to distinguish merged plaques and consider

them as single one. Besides, I was wondering how statistics were performed since the significance between controls and target groups are always too high. For statistical analysis, I suggest to take the average of plaque sizes from each assay and process with only this values from independent assays to run student's t-test.

3. When rPKG kinase activity was tested, at about 2-3% ATP conversion was observed in the absence of cGMP as given in the Fig 3c. Could that be possible that NcPKG has a minor but negligible other kinase activity by using ATP or is it just a background from kit measurement? There were also several bands in the representative western blot image of purified rPKG protein given at Supp. Fig 2h. The kinase activity shown in the Fig 3c may not only correspond to rPKG protein, but also includes the activity of another bacterial protein. I am sceptical about the purity of the processed protein content. I would suggest to use a real negative assay control as "heat-inactivated/degraded rPKG protein" and set the assay accordingly. This will eliminate confusions and be more supportive of given outcomes.

4. Possession of ion transport domain in the protein sequence and in silico prediction of the protein structure would not be enough to focus only calcium ion as the target of the NCLIV_005460, so-called calcium channel-associated protein (NcCACNAP) by the authors. Although the function of this protein during the process of calcium influx has been evidently shown by the authors, NCLIV_005460 channel can also responsible for Na⁺ or K⁺ transportation (influx or efflux) at the same time as it was predicted to be a member of cation channel transporter family. It is also known that there are several transporters using different ligands than transported ions, such as K⁺-Dependent Na⁺/Ca²⁺-Exchangers or Ca²⁺ activated K⁺ channels. Can authors do additional assays to test the functionality of NCLIV_005460 channel on sodium and potassium fluctuation? Since the role of both potassium and calcium has already been demonstrated on the successful egress of parasite *T. gondii* (PMID: 33524795), it would be good to know whether *N. caninum* follows the same path to egress from host cell.

5. In Fig 3f, proteins which are detected as downstream targets of NcPKG and associated with ion transport and metabolism were shown in a heat map. In the text, the number of significantly regulated aforesaid proteins were mentioned as 19; however, there are more than 19 in Fig 3f. I suggest to highlight the emphasized 19 proteins in the figure as a complementation of the text part.

6. It is not clear to me why transcript level in CACNAP-S243D mutant significantly increased when compared with unmutated, complementation mutants as well as dephosphorylated (S243A) mutants. It was explained by the authors that 243S site may be associated with the transcriptional regulation of CACNAP, and the phosphorylation of this site may promote gene transcription. Is there an indication to support this idea? In silico analysis of the closest orthologue of CACNAP, or an evaluation of a member of T-type voltage-gated Ca²⁺ channel may help for better understanding. The notion should be supported by a publication.

Minor Comments:

1. Line 130: Please give predicted molecular weight of PKG in kDa rather than Da. It seems a bit odd.

2. Line 201: An additional subtitle can be used here as the effect of MBP164-78 on the inhibition of NcPKG has been described in the first paragraph until the line 220. After that, quantitative phosphoproteome results were given, which will be suitable to sum under the title of "NcPKG influences various signaling pathways in *N. caninum* as revealed by quantitative phosphoproteome". Concurrently, Figure 3a, b and c can be split from 3d,e,f and given with a separate figure legend.

3. Lines 207-209: "Here, *N. caninum* tachyzoites treated with the inhibitor MBP164-78 (Fig. 3a) were detected, and the inhibition of kinase activity of recombinant protein rNcPKG was analyzed". I assume the impact of MBP164-78 on *N. caninum* tachyzoites were detected? The sentences must be corrected!

4. Supplementary Figure 1b: Gel image confirms successful 5' and 3' - homologous recombination of GCaMP6f by 4 different primer pairs. I guess numbers given under each pair represents Transgenic and its control parental strain with numbers 1 and 2, which is confusing since there is no explanation in the figure legend. I suggest to use the letters "T" and "P" which stands for transgenic and parental strains, respectively under the each stated PCR condition. Please give the statement in the legend.

5. Supplementary Figure 2h is irrelevant with the figure legend. Please separate it and combine with the Fig 3a, b and c for better combination.

6. Organism names in abbreviations; such as Nc in NcPKG or Pf in PfSUB1 must be given as italic. Please correct them through the manuscript.

7. Fig Legend 1. Line 1069: "Western blotting images showed that the protein was almost completely degraded after 8 h." Since there are traces of both bands in Western Blot image, and IFA still shows the residual PKG-Ty staining, I suggest not to use "completely degraded" as a term. The expression should be softened.

8. There are excessive numbers of grammatical errors in the manuscript. Please go over carefully to correct them.

Version 1:

Reviewer comments:

Reviewer #1

(Remarks to the Author)

Although the manuscript has improved, the claim that PKG functions through phosphorylation of CACNAP at Serine 243 is still not backed by experimental data.

The authors were still not able to complement the CACNAP knockout phenotype even when using the native promoter of CACNAP (rebuttal letter). It is difficult to understand why they were able to generate complemented strains with a tubulin promoter but were unable to complement the knockout parasites with the endogenous CACNAP promoter driving wild-type CACNAP or CACNAP with the S243 mutation expression. I understand that generating these strains and redoing some of the work takes effort but I am currently not convinced that the CACNAP phenotype is related to the PKG phenotype. However, the authors now acknowledge that more work needs to be done to demonstrate the connection between PKG and CACNAP.

Reviewer #2

(Remarks to the Author)

I believe that the manuscript is improved following the changes made in response to the three reviewers.

Some specific additional changes are needed:

The authors still refer to 'cGMP-dependent protein kinase G' in the abstract. It should be 'cGMP-dependent protein kinase (PKG)'. It is correct on lines 53-54.

Line 155 does not make sense and should be rewritten: '...PKG's effect of PKG on....'

Line 344-5 I think this first sentence which mentions 'unanticipated plasticity' should be rewritten as the meaning is not clear.

Line 346, it should be clarified that phosphodiesterases hydrolyse cyclic nucleotides

Line 466 Rep;ace 'not quite sure' with e.g. 'are inconclusive'

Reviewer #3

(Remarks to the Author)

Thanks to the authors for taking my comments into account and making careful corrections on the manuscript. Considerable work has been done to improve the quality of the work.

Minor further Comments:

Please correct the spelling of IP3 as IP₃ throughout the manuscript.

Please correct the spelling of PKAC1 as PKAc1 throughout the manuscript.

Line 500_ Please add the description as follows into the methods part under "Parasites and host cells culture" section: "Tubulin promoter-driven OstTIR1 sequence (Table S1) was inserted into the non-coding region between NCLIV-058880 and NCLIV-058890 genes using the CRISPR-Cas9 system to generate NcTir1 parasite strain".

Line 470_ Please give the reference to the statement given as:

"It is also known that there are several transporters using different ligands than transported ions, such as K⁺-471 Dependent Na⁺/Ca²⁺-Exchangers or Ca²⁺ activated K⁺ channels"

Line 719_ buffer - "r" is missing.

Reviewer #1	
Comments	Response
Main Comments: 1) The authors were not able to complement the CACNAP knockout phenotype (Figure 6). They speculate that this was because of overexpression from the tubulin promoter. However, they should demonstrate that the CACNAP knockout phenotypes were indeed due to CACNAP deletion, and not to an off-target effect, by complementing with a CACNAP construct using the endogenous promoter. They should also show the role of the S243 phosphorylation by complementing the knockout with an S243 mutant expressed from the endogenous promoter. Currently the connection between PKG and CACNAP with regards the phenotypes observed is speculative as it was not shown that the S243 phosphorylation is important. 2) The authors claim that the S243D mutations lead to an increase in transcript expression (Fig 6e). However, it seems very unlikely that this amino acid change could lead to transcript level changes. It is more likely that when they complemented the knockout strain multiple copies of the complementation construct were inserted into the parasite genome. Minor comments: There were multiple instances of sentences that were not very clear. For example: 1) Line 76 resistance in door-spread parasites. What are door-spread parasites? 2) Lines 156-157 Analyzing the effect of PKG on the natural egress process, the tachyzoite of the PKG-mAID strain was simultaneously inoculated with multiple cell climbs. It is unclear what the authors mean by this (multiple cell climbs?) 3) Line 188 genetically encoded Ca²⁺-indicating manner -> Ca²⁺-indicator 4) 205 Previous studies have d that -> have	Main Comments: 1) We would like to express our sincere gratitude for your professional comments on our article. As you have pointed out, CACNAP- complement strains we constructed were unable to fully rescue the CACNAP knockout phenotype, likely due to the high expression level of CACNAP. Based on your suggestion, we attempted to construct a rescue strain and an S243 mutant strain using the CACNAP promoter within the CACNAP knockout strain. Despite multiple efforts, we were unfortunately unable to obtain these specific strains. To strengthen the credibility of the CACNAP knockout phenotype, we repeated the main phenotypic experiments using two different monoclonal strains (as shown in Supplementary Figure 5f and 5g). The consistency of the results confirms that the phenotype is indeed attributable to the absence of CACNAP, rather than off-target effects. Additionally, we acknowledge the limitations in elucidating the precise relationship between PKG and CACNAP. What we currently understand is that inhibiting PKG results in low 243S phosphorylation of CACNAP in parasites. We have clarified this point in the discussion section (line 452-462) to address potential questions on this interaction. 2) We sincerely thank you for your valuable feedback. We had noticed this phenomenon but were initially unable to explain the result. After careful consideration, we decided to remove the subtitle and instead summarize this finding in a Supplementary Figure 6. Importantly, this adjustment dose not impact the overall conclusion of the article. Minor comments:

demonstrated that 5) The Toxoplasma version of NCLIV_005460 was endotagged and described in J Eukaryot Microbiol. 2018 Jul; 65(5): 709–721. doi: 10.1111/jeu.12626, which the authors could cite. 6) 344 Science we have previously shown that -> Since we have previously shown that	We sincerely apologize for the oversights in our manuscript and appreciate your attention to detail. Based on your comments, we have made the following corrections to ensure consistency throughout the text: 1) We have corrected the “door-spread parasites” into “apicomplexan parasites” in line 75. 2) Removed "with multiple cell climbs" to improve sentence flow in line 156. 3) Revised “genetically encoded Ca²⁺-indicating manner” to “a genetically-encoded calcium sensor” in line 183-184. 4) Corrected “have d that” to “have demonstrated that” in line 202. 5) Added the appropriate citation as suggested in line 252 of the revised manuscript. 6) Corrected “Science we have previously shown that” to “Since we have previously shown that” .
---	---

Reviewer #2

Comments	Response
Introduction, 1. Line 52, I would refer to the protein as the ‘cGMP-dependent protein kinase’ and then ‘PKG’ thereafter. It is also known as ‘protein kinase G’. Not ‘cGMP-dependent protein kinase G’. 2. Line 54, I presume this should be the ‘γ-phosphate group’. 3. Line 58-60, should there be a reference for the cGKI information? Perhaps all is within reference 8. 4. Line 76, is ‘door spread parasites’ intended? If so, reference 11 is not an appropriate citation. 5. PMID: 33762339 could be cited in	Thank you so much for your thorough review. We have made the following corrections to the manuscript based on your suggestions: Introduction 1. Corrected “cGMP-dependent protein kinase G” to “cGMP-dependent protein kinase r” in line 54. 2. Corrected “γ-phosphorylated subunit” into “γ-phosphate group” in line 56. 3. Added a reference about cGKI in line 56. 4. Corrected “door spread parasites” into “apicomplexan parasite” in line 75. 5. Cited the literature (PMID: 33762339) as suggested in line 87. 6. Corrected “intracellular” to “apicomplexan” in line 104. Results

relation to the involvement of PKG and calcium in apicomplexan parasites. 6. Line 106, perhaps replace ‘intracellular’ with ‘apicomplexan’. Results 7. Line 119, is it homology (BLAST) searching rather than secondary structure analysis that revealed this? 8. Lines 120-21, is it clear from the sequence data that all four CAP_ED domains are likely to be functional cGMP-binding domains, or only three of the four domains as is the case in e.g. P. falciparum where key conserved amino acids are absent from site C? Lines 122-24, should be reworded. 9. Lines 128-30, only later in the discussion the significance of these two bands with respect to coccidian dual N-terminal (dual) acylation of PKG. I think it should be also mentioned briefly here. 10. Line 157, is ‘cell climbs’ intended? Cell cycles perhaps? 11. Line 167, this should be ‘phosphodiesterases’ rather than ‘phosphodiester acid’. To my knowledge, the reason that 8-Br-cGMP is effective is that it is one of two cGMP analogues that can bind apicomplexan PKG effectively (rather than the rate of hydrolysis). This was previously shown by e.g. Robert Donald and colleagues (PMID: 11914085). 12. Line 188, could this be changed to e.g. ‘a genetically-encoded calcium sensor’? 13. Line 205, presumably d= demonstrated.	7. Corrected “secondary structure analysis that revealed” to “homology (BLAST) searching revealed” in line 116-117. 8. We apologize for the lack of rigor in our initial description. Our intention was to convey that PKG contains four CAP_ED domains that interact with cGMP. However, the primary functional domain among these has not been identified or tested, and it remains unclear if their roles are consistent with those in P. falciparum. Therefore, we have revised lines 120-124 as follows: "Homology research (BLAST) revealed that NcPKG contains four tandem copies of the cyclic nucleotide-binding domain (CAP family effector domains, CAP_ED) and a catalytic domain for serine/threonine kinases (STKc_cGK) (Fig. 1a), which catalyzes the transfer of the γ-phosphoryl group from ATP to serine/threonine residues on protein substrates." in line 116-120. 9. We have added an analysis of the two bands of PKG in line 128-131. The supplementary content states: "The two isoforms of NcPKG may be expressed by alternative translation initiation, similar to TgPKG-I, which localizes to the plasma membrane via N-acylation, governing PKG function and governs PKG function. In contrast, the smaller protein isoform, TgPKG-II is cytosolic due to the absence of N-terminal acylation residues." 10. We have removed "with multiple cell climbs" to enhance the natural flow of the sentence in line 156. 11. We have removed " phosphodiester acid " to enhance the natural flow of the sentence. As your suggestion, we have revised the description of 8-Br-cGMP to “a cell membrane permeable cGMP derivative that can bind apicomplexan PKG effectively” and cited the literature (PMID: 11914085) in line 164-166.
---	---

14. Lines 210-11, consider changing: ‘indicating the sensitivity of MBP146-78 to the inhibitory activity of N. caninum’ to e.g. ‘indicating the sensitivity of N. caninum PKG to MBP146-78’. 15. Figure 3f, these data would be much more accessible to readers if the proteins (where annotations are available) are actually named to the left (rather than having NCLIV numbers). This would also allow readers to quickly identify which row corresponds to the putative calcium channel if it is labelled NcCACNAP and other proteins of interest. The text in Figure 3e and Supplementary Figure 3b is too small and should be enlarged. 16. Figure 6 In the legend to Figure 6c and e, ‘com’ should be defined. I assume that it is complementation with the wild type sequence. 17. Line 329, I assume this refers to complementation of the knockout. State that complementation with one of two different mutations were performed in parallel. 18. Line 343. I assume that ‘normal strain’ means wild type? 19. Line 344, change ‘science’ to since. 20. Figure 6e, the RNA expression levels are very low in Nc1 (I assume this is wild type) and zero in the knockout (as expected, also, note that the y-axis of Figure 6e has no units). 21. Lines 346-55, I am confused as to whether the authors are saying that 1) the elevated transcription levels may be an artifact of the strong tubulin promoter used in the complementation strategy (and the consequent calcium overload), or 2) that this phosphorylation site could be involved in	12. We have corrected “genetically encoded Ca²⁺-indicating manner” to “a genetically-encoded calcium sensor” in line 184. 13. We have corrected “have d that” to “have demonstrated that” in line 202, 14. We have corrected Lines 210-211 to “Similarly, N. caninum tachyzoites hardly formed plaques when incubated with MBP146-78 (Fig. 3b), indicating the sensitivity of N. caninum PKG to MBP146-78.” in line 204-206. 15. In Figure 3f (Figure 3d of the newly submitted manuscript), we have corrected “NCLIV_005460” to “CACNAP”. The remaining proteins are presumed to be proteins in N. caninum and their functions are currently unknown. Since most of these proteins lack reported names, we believe that using “NCLIV numbers” provides a more accurate reference to facilitate related research. Additionally, we have enlarged the text in Figure 3e (Figure 3c of the newly submitted manuscript) and Supplemented Figure 3b (Supplemented Figure 3d of the newly submitted manuscript). 16. In the legends for Figure 6c and e (Supplemented Figure 6 of the newly submitted manuscript), we have clarified that ‘com’ represents the strain that supplemented the CACNAP sequence in CACNAP knockout strain, now corrected to “Δcacnap/CACNAP”. Additionally, the strain supplemented with the CACNAP S243A mutant sequence is referred to as “Δcacnap/CACNAP^{S243A}”, and the strain supplemented with the CACNAP S243D mutant sequence is labeled as “Δcacnap/CACNAP^{S243D}”. 17. We have supplemented the CACNAP sequence with the quasi-phosphorylated S243D sequence and the dephosphorylated S243A mutation sequence in the knockout strain, respectively in line 315-317.
--	--

transcriptional regulation of the gene (especially since it is within a motif predicted to be involved in transcriptional regulation). The position of the authors on these two possibilities needs to be clarified. Also some explanation of why the phosphomimetic (S243D) apparently leads to the highest levels of RNA expression. The subtitle of this section I think needs re-writing to reflect the results of these experiments (see below)*. The current subtitle I believe is an inaccurate reflection of the findings. Particularly since the S243A mutant (which will not be phosphorylated) also has a very high level of RNA expression.

Whilst the findings are interesting, I don't think any conclusions can be drawn from these experiments regarding the role of this phosphorylation site in protein function. Therefore I disagree with the statement on lines 366-7. Perhaps the use of an alternative promoter (more appropriate than tubulin) in the complementation system would be a good strategy going forward. *In my opinion it would be a good idea to totally omit this section of the results from the paper and perhaps only mention very briefly in the Discussion that this was attempted, but that the results were inconclusive, and why. I believe there is no mention of these findings in the Abstract. How do the authors envisage this protein interacting with the transcriptional machinery given its location in the cell.

Discussion

22. Line 372, insert: ...only 'one' gene encoding....
23. Line 373, as mentioned earlier, no analysis is presented regarding whether the four potential cGMP-binding domains of NcPKG are likely to be functional. In other apicomplexans there are four adjacent

18. We have corrected "normal strain" to "wild type" in line 327.

19. We have corrected "science" to "since".

20. We have changed the Y-axis label to "Relative mRNA of CACNAP" in Figure 6e (Supplemented Figure 6e of the newly submitted manuscript).

21. We appreciate this suggestion. Initially, we found observed that supplementation with CACNAP did not rescue the phenotype caused by the loss of CACNAP; in fact, the damage was even worse. Subsequently, we found that the mRNA expression level of supplemented CACNAP was significantly higher than the endogenous expression in wild-type strains, likely due to overexpression driven by strong tubulin promoters. However, we have insufficient evidence to determine whether phosphorylation sites affect transcriptional expression, and we could not find relevant literature to explain this phenomenon. Consequently, we have deleted the corresponding content and rewritten this part according to the reviewer's suggestion.

Discussion

22. We have deleted this sentence from the discussion.

23. We have rewritten the text in the revised manuscript.

24. We have corrected "normal" to "wild type" in line 327.

25. We have removed this description from the discussion.

extended sequence motifs, three of which are functional and a fourth one lacks some key residues required for cGMP binding. The statement here implies that all four in Nc look to be functional. This needs clarification and justification here and earlier. The Discussion reiterates the results too extensively as evidenced by citing the figures throughout. 24. Line 412, the term ‘normal group’ needs to be changed. 25. Lines 475-477, as mentioned previously, I don’t think the statement here is supported by the data.	
Reviewer #3	
Comments	Response
Major Comments: 1. It is not clear how auxin sensitive TIR1 stain was generated. Lines 390-392 “The codon-optimized auxin receptor Tir1 was expressed in the cytoplasm of Nc1 strain (originally cloned by Yang, preserved at China, Agricultural University, Beijing)” For the reproducibility of the work by the others, it is essential to mention which region has been targeted in the genome to express TIR1 receptor, or how it was expressed in the cytosol. If the strain was developed using the same strategy as recently published by Mineo et al. (PMID: 35019667), then this paper should be cited properly. 2. In Fig 1f, Fig 5a and Fig 6d the analysis of the plaque area must be performed again using single plaque images and processing with the size measurements using ImageJ, as Photoshop unable to distinguish merged plaques and consider them as single one. Besides, I was wondering how statistics	1. The Nc1-Tir1 strain was generated in our laboratory (Yang, et.al), with the Tubulin promoter-driven OsTIR1 sequence inserted into the non-coding region between the NCLIV-058880 and NCLIV-058890 genes using the CRISPR-Cas9 system. At the same time of generating and applying this Nc1-Tir1 strain, the study by Mineo et al. (PMID: 35019667) had not yet been published, which it is not cited in our work. To facilitate reproducibility for other researchers, we have published the OsTIR1 expression sequence in Supplement Table 1. 2. For plaque analysis, plaque areas were measured in pixel points using Photoshop C6S software (Adobe, USA). The data were collected from three independent experiments with plaque areas analyzed using an unpaired student’s t-test. On the basis of your suggestion, we have re-conducted the statistical analysis on Fig 1f, Fig 5a and Fig 6d (Supplemented Figure 6d of the newly submitted manuscript). We calculated the average plaque size from each assay and used these averaged values

were performed since the significance between controls and target groups are always too high. For statistical analysis, I suggest to take the average of plaque sizes from each assay and process with only this values from independent assays to run student's t-test.

3. When rPKG kinase activity was tested, at about 2-3% ATP conversion was observed in the absence of cGMP as given in the Fig 3c. Could that be possible that NcPKG has a minor but negligible other kinase activity by using ATP or is it just a background from kit measurement? There were also several bands in the representative western blot image of purified rPKG protein given at Supp. Fig 2h. The kinase activity shown in the Fig 3c may not only correspond to rPKG protein, but also includes the activity of another bacterial protein. I am sceptical about the purity of the processed protein content. I would suggest to use a real negative assay control as "heat-inactivated/degraded rPKG protein" and set the assay accordingly. This will eliminate confusions and be more supportive of given outcomes.

4. Possession of ion transport domain in the protein sequence and in silico prediction of the protein structure would not be enough to focus only calcium ion as the target of the NCLIV_005460, so-called calcium channel-associated protein (NcCACNAP) by the authors. Although the function of this protein during the process of calcium influx has been evidently shown by the authors, NCLIV_005460 channel can also responsible for Na⁺ or K⁺ transportation (influx or efflux) at the same time as it was predicted to be a member of cation channel transporter family. It is also known that there are several transporters using different ligands than transported ions, such as K⁺-

from the independent assays to perform the student's t-test.

3. We sincerely appreciate your valuable comments. In fact, we had considered this aspect during the enzyme activity tests. The negative control used was indeed heat-inactivated rPKG protein, which yielded results comparable to the blank control (data not shown). We will emphasize this detail in our methods section. We believe that the 2-3% ATP conversion observed in the absence of cGMP is more likely due to degradation of the recombinant PKG protein also seen in multiple bands in western blot of Supplementary Figure 2h (Supplemented Figure 2j of the newly submitted manuscript). We speculate that the structural integrity of PKG may be destroyed, but the catalytic activity of the kinase is retained, and therefore the ATP can be catalyzed in the absence of cGMP. Nevertheless, we believe this minor ATP conversion does not significantly impact the overall conclusions of our experiment.

4. Thank you for your insightful comments. Indeed, our results confirm only that CACNAP can transport Ca²⁺, which is why we designated it as a Ca²⁺ channel-associated protein rather than a calcium channel protein. The possibility of CACNAP transport Na⁺ or K⁺ could be analyzed by patch-clamp experiments. However, currently, no experimental protocols are available for this analysis in *N. caninum*, and our laboratory lacks the equipment for patch-clamp analysis. Therefore, we are unable to determine any potential correlation between CACNAP and potassium ions. We have incorporated this point into our discussion in line 455-462.

5. We identified 19 proteins as downstream targets of NcPKG, associated with ion transport and metabolism. The heat map shows changes in the phosphorylation sites

Dependent Na⁺/Ca²⁺-Exchangers or Ca²⁺ activated K⁺ channels. Can authors do additional assays to test the functionality of NCLIV_005460 channel on sodium and potassium fluctuation? Since the role of both potassium and calcium has already been demonstrated on the successful egress of parasite *T. gondii* (PMID: 33524795), it would be good to know whether *N. caninum* follows the same path to egress from host cell.

5. In Fig 3f, proteins which are detected as downstream targets of NcPKG and associated with ion transport and metabolism were shown in a heat map. In the text, the number of significantly regulated aforesaid proteins were mentioned as 19; however, there are more than 19 in Fig 3f. I suggest to highlight the emphasized 19 proteins in the figure as a complementation of the text part.

6. It is not clear to me why transcript level in CACNAP-S243D mutant significantly increased when compared with unmutated, complementation mutants as well as dephosphorylated (S243A) mutants. It was explained by the authors that 243S site may be associated with the transcriptional regulation of CACNAP, and the phosphorylation of this site may promote gene transcription. Is there an indication to support this idea? In silico analysis of the closest orthologue of CACNAP, or an evaluation of a member of T-type voltage-gated Ca²⁺ channel may help for better understanding. The notion should be supported by a publication.

Minor Comments:

1. Line 130: Please give predicted molecular weight of PKG in kDa rather than Da. It seems a bit odd.

of these proteins. We have updated “NCLIV_005460” to “CACNAP” in Figure 3f (Figure 3d of the newly submitted manuscript). The remaining proteins are presumed *N. caninum* proteins with unknown functions. Since most of these proteins lack reported names, we believe that using “NCLIV numbers” provides a more precise reference, which will facilitate future research on these targets.

6. According to the reviewer's suggestion, we have removed this section from the results. This deletion is not affected the overall conclusion of the article.

Minor Comments:

1. We have corrected “Da” to “kDa” in line 128.

2. We have used the title “NcPKG influences various signaling pathways in *N. caninum* as revealed by quantitative phosphoproteome” in line 198-199.

Figure 3a, b and c have been split from 3d, e, and f and given with a separate figure legend. Figure 3a, b, and c have been moved to supplemental Figure 2.

3. We have revised Lines 207-209 to “Similarly, *N. caninum* tachyzoites hardly formed plaques when incubated with MBP146-78 (Fig. 3b), indicating the sensitivity of *N. caninum* PKG to MBP146-78.” in line 204-206.

4. We have modified the annotation in Supplementary Figure 1 to clarify that GCaMP6F refers to transgenic strains, while Nc1 denotes the parental strains.

5. Figure 3a, b, and c have been moved to Supplemental Figure 2 and reordered accordingly.

Additionally, Supplementary Figure 2h is related to Figure 3c (Supplemental Figure 2h of the newly submitted manuscript), which we believe should be retained and reordered as well.

2. Line 201: An additional subtitle can be used here as the effect of MBP164-78 on the inhibition of NcPKG has been described in the first paragraph until the line 220. After that, quantitative phosphoproteome results were given, which will be suitable to sum under the title of “NcPKG influences various signaling pathways in *N. caninum* as revealed by quantitative phosphoproteome”. Concurrently, Figure 3a, b and c can be split from 3d,e,f and given with a separate figure legend.

3. Lines 207-209: “Here, *N. caninum* tachyzoites treated with the inhibitor MBP164-78 (Fig. 3a) were detected, and the inhibition of kinase activity of recombinant protein rNcPKG was analyzed”. I assume the impact of MBP164-78 on *N. caninum* tachyzoites were detected? The sentences must be corrected!

4. Supplementary Figure 1b: Gel image confirms successful 5' and 3'- homologous recombination of GCaMP6f by 4 different primer pairs. I guess numbers given under each pair represents Transgenic and its control parental strain with numbers 1 and 2, which is confusing since there is no explanation in the figure legend. I suggest to use the letters “T” and “P” which stands for transgenic and parental strains, respectively under the each stated PCR condition. Please give the statement in the legend.

5. Supplementary Figure 2h is irrelevant with the figure legend. Please separate it and combine with the Fig 3a, b and c for better combination.

6. Organism names in abbreviations; such as Nc in NcPKG or Pf in PfSUB1 must be given as italic. Please correct them through

6. We have corrected the abbreviations in the manuscript.

7. We have revised the statement “Western blotting images showed that the protein was almost completely degraded after 8 h” to “Western blotting images showed that the protein can't be detected after 8 h” in line 1035.

8. Thanks for your suggestion. We will carefully review the entire manuscript to correct all issues. Your feedback is invaluable for improving the quality of our work, and we will take it seriously

the manuscript. 7. Fig Legend 1. Line 1069: “Western blotting images showed that the protein was almost completely degraded after 8 h.” Since there are traces of both bands in Western Blot image, and IFA still shows the residual PKG-Ty staining, I suggest not to use “completely degraded” as a term. The expression should be softened. 8. There are excessive numbers of grammatical errors in the manuscript. Please go over carefully to correct them.	
Comments	Response
We have noted that citation of previous cGMP signaling work in T. gondii is highly selective. For example, previously reported PKG, PDE and P4-ATPase-GC works have not been properly cited and discussed.	Thanks for your suggestions. The references you suggested are benefit for us. We have read this and learn a lot. Some papers play a key role in improving the quality of article. Hence, we have cited them in line 51-53, line362-379 in the revised manuscript. Thank you again.

Reviewer #1	
Comments	Response
Although the manuscript has improved, the claim that PKG functions through phosphorylation of CACNAP at Serine 243 is still not backed by experimental data. The authors were still not able to complement the CACNAP knockout phenotype even when using the native promoter of CACNAP (rebuttal letter). It is difficult to understand why they were able to generate complemented strains with a tubulin promoter but were unable to complement the knockout parasites with the endogenous CACNAP promoter driving wild-type CACNAP or CACNAP with the S243 mutation expression. I understand that generating these strains and redoing some of the work takes effort but I am currently not convinced that the CACNAP phenotype is related to the PKG phenotype. However, the authors now acknowledge that more work needs to be done to demonstrate the connection between PKG and CACNAP.	We sincerely thank you for your valuable feedback. Having taken your comments into serious consideration, in the newly revised manuscript, we have deleted the statement that CACNAP is the downstream protein of PKG. Also, in the Results section, we have removed the related content about the connection between PKG and CACNAP in line 330-334. Moreover, we have explained the limitations of this paper in the discussion part in line 458-468.
Reviewer #2	
Comments	Response
I believe that the manuscript is improved following the changes made in response to the three reviewers. Some specific additional changes are needed:  1. The authors still refer to ‘cGMP-dependent protein kinase G’ in the abstract. It should be ‘cGMP-dependent protein kinase (PKG)’. It is correct on lines 53-54. 2. Line 155 does not make sense and should be rewritten: ‘...PKG’s effect of PKG on....’ 3. Line 344-5 I think this first sentence which mentions ‘unanticipated plasticity’ should be rewritten as the meaning is not clear. 4. Line 346, it should be clarified that 	We sincerely apologize for the oversights in our manuscript and appreciate your attention to detail. Based on your comments, we have made the following corrections to ensure consistency throughout the text:  1. In the newly revised manuscript, we corrected “cGMP-dependent protein kinase G” to “cGMP-dependent protein kinase (PKG)” in line 14. 2. Revised the sentence in line 153-158. 3. Rewrote the sentence in line 336-337. 4. Rewrote the sentence line 334-336. 5. Corrected “not quite sure” to “inconclusive” in line 471.

phosphodiesterases hydrolyse cyclic nucleotides 5. Line 466 Replace ‘not quite sure’ with e.g. ‘are inconclusive’	
Reviewer #3	
Comments	Response
Thanks to the authors for taking my comments into account and making careful corrections on the manuscript. Considerable work has been done to improve the quality of the work. Minor further Comments:  1. Please correct the spelling of IP3 as IP₃ throughout the manuscript. 2. Please correct the spelling of PKAC1 as PKAc1 throughout the manuscript. 3. Line 500_ Please add the description as follows into the methods part under “Parasites and host cells culture” section: “Tubulin promoter-driven OsTIR1 sequence (Table S1) was inserted into the non-coding region between NCLIV-058880 and NCLIV-058890 genes using the CRISPR-Cas9 system to generate NcTir1 parasite strain”. 4. Line 470_ Please give the reference to the statement given as: “It is also known that there 470 are several transporters using different ligands than transported ions, such as K⁺-471 Dependent Na⁺/Ca²⁺-Exchangers or Ca²⁺ activated K⁺ channels” 5. Line 719_ buffer - “r” is missing. 	Thank you very much for your thorough review. We have made the following corrections to the manuscript based on your suggestions:  1. In the newly revised manuscript, we corrected “IP3” to “IP₃”. 2. Corrected “PKAC1” to “PKAc1”. 3. We added the following description to the Methods section in line 505-507. 4. As your suggestion, we have supplemented the references in line 476-478. 5. Corrected “bufe” to “buffer” in line 727.
Comments	Response
As the manuscript mostly confirms findings in Toxoplasma, the omitted relevant literature on cyclic nucleotide signaling in Toxoplasma must be included in a revised bibliography.	Thank you for your feedback. We acknowledge the importance of the relevant literature on cyclic nucleotide signaling in Toxoplasma. We'll conduct a literature search and include all the omitted studies in the revised bibliography.